# Astrocytic p38α MAPK drives NMDA receptor-dependent long-term depression and modulates long-term memory

Marta Navarrete [1,2], María I. Cuartero [1], Rocío Palenzuela[1,3], Jonathan E. Draffin[1], Ainoa Konomi[1], Irene Serra[2], Sandra Colié[4], Sergio Castaño-Castaño[5], Mazahir T. Hasan[5,6], Ángel R. Nebreda[4,7] & José A. Esteban [1]

NMDA receptor-dependent long-term depression (LTD) in the hippocampus is a well-known form of synaptic plasticity that has been linked to different cognitive functions. The core mechanism for this form of plasticity is thought to be entirely neuronal. However, we now demonstrate that astrocytic activity drives LTD at CA3-CA1 synapses. We have found that LTD induction enhances astrocyte-to-neuron communication mediated by glutamate, and that $Ca^{2+}$ signaling and SNARE-dependent vesicular release from the astrocyte are required for LTD expression. In addition, using optogenetic techniques, we show that low-frequency astrocytic activation, in the absence of presynaptic activity, is sufficient to induce post-synaptic AMPA receptor removal and LTD expression. Using cell-type-specific gene deletion, we show that astrocytic p38α MAPK is required for the increased astrocytic glutamate release and astrocyte-to-neuron communication during low-frequency stimulation. Accordingly, removal of astrocytic (but not neuronal) p38α abolishes LTD expression. Finally, this mechanism modulates long-term memory in vivo.

[1] Department of Neurobiology, Centro de Biología Molecular "Severo Ochoa" (CSIC-UAM), 28049 Madrid, Spain. [2] Instituto Cajal (CSIC), 28002 Madrid, Spain. [3] Faculty of Experimental Sciences, Universidad Francisco de Vitoria, Pozuelo de Alarcón, 28223 Madrid, Spain. [4] Institute for Research in Biomedicine (IRB Barcelona), Barcelona Institute of Science and Technology, 08028 Barcelona, Spain. [5] Achucarro Basque Center for Neuroscience, 48940 Leioa, Spain. [6] Ikerbasque–Basque Foundation for Science, 48013 Bilbao, Spain. [7] ICREA, Pg. Lluís Companys, 08010 Barcelona, Spain. Correspondence and requests for materials should be addressed to M.N. (email: mllinas@cajal.csic.es) or to J.A.E. (email: jaesteban@cbm.csic.es)

Hippocampal long-term depression (LTD), a long-lasting decrease of synaptic transmission strength, is induced by prolonged periods of low-frequency stimulation (LFS)[1]. Mounting evidence has revealed the relevance of LTD for some forms of learning and memory encoding, as well as for cognitive effects of stress and drug addiction[2]. Much effort in the field has been directed toward understanding the underlying mechanisms that account for the change in synaptic strength. In the case of N-methyl-D-aspartate receptor (NMDAR)-dependent LTD, even though the molecular details are still being elucidated, it is well accepted that the core mechanism relies exclusively on neuronal events: (i) prolonged, low-frequency release of glutamate from the presynaptic terminal, (ii) activation of postsynaptic NMDARs, (iii) engagement of specific signaling cascades at the postsynaptic terminal, and (iv) endocytic removal of α-amino-3-hydroxy-5-methyl-4-isoxazolepropionic acid receptors (AMPARs) from the postsynaptic membrane[3]. However, despite a large number of publications on this topic, the potential participation of glial cells in this sequence of events has received little attention.

Increasing evidence has indicated that astrocytes play a much more active role in brain physiology than previously considered, including processing of synaptic information[4]. Specifically, it is generally accepted that astrocytes may directly modify synaptic activity by the release of active molecules, called gliotransmitters[5], although this issue is still under intense debate[6,7]. One of the most common gliotransmitters is glutamate[8], whose release from astrocytes in response to endocannabinoids leads to the facilitation[9] or inhibition[10] of glutamate release from the presynaptic neuronal terminal, or the activation of NMDARs at the postsynaptic terminal[11]. However, it is not known whether and how glutamate released by astrocytes may contribute to standard NMDAR-dependent LTD induced by LFS.

In this report, we show that astrocytes are core elements of the LTD mechanism by providing the glutamate transmitter responsible for postsynaptic depression of CA3–CA1 synaptic transmission. This mechanism requires $Ca^{2+}$ signaling and SNARE-dependent exocytosis in the astrocyte. In addition, we show that the activity of p38α mitogen-activated protein kinase (MAPK) in the astrocyte (and not in the neuron) is required for hippocampal LTD, and modulates long-term memory in vivo.

## Results

**LTD induction evokes neuron–astrocyte communication.** We first investigated whether conventional LTD induced by low-frequency stimulation (LFS; 1 Hz, 300 pulses) of Schaffer collaterals (SCs) triggers astrocytic activity. To this end, we monitored $Ca^{2+}$ levels in astrocytes located in the *stratum radiatum* of acute hippocampal slices by two-photon laser-scanning fluorescence microscopy using a genetically encoded $Ca^{2+}$ indicator (GCaMP6f). Expression was specifically directed to astrocytes by in vivo injections of adeno-associated viruses (AAV2/5) bearing the GFAP promoter (Fig. 1a, b). The LFS protocol evoked $Ca^{2+}$ elevations in 380 out of 440 recorded astrocytes ($n = 20$ slices) (Fig. 1b, c). The frequency of these $Ca^{2+}$ signals was dependent on the distance to the LFS, being observable 150–200 μm away from the stimulation electrode (Supplementary Fig. 1a, b). Similar results were obtained with fluo-4-loaded hippocampal slices (63 out of 79 recorded astrocytes, $n = 6$ slices) (Supplementary Fig. 1c, d).

Accumulating data from different laboratories indicate that astrocyte $Ca^{2+}$ signals elicit astrocyte–neuron communication in the form of neuronal slow inward currents (SICs) mediated by glutamate released from astrocytes and activation of neuronal NMDARs[12–15]. We then examined whether LFS of SCs elicits glutamate-mediated SICs in CA1 neurons. Indeed, the LFS protocol caused an increase in the frequency of SICs (Fig. 1d), supporting the interpretation that an increase in astrocytic intracellular $Ca^{2+}$ concentration during LTD induction is accompanied by glutamate release from astrocytes. To note, even if SICs are relatively infrequent (compared to 1 Hz EPSCs during LFS), their charge transfer is considerably larger (75-fold, on average), because of their much longer duration (Supplementary Fig. 2). As control, incubation with the NMDAR blocker AP5, virtually abolished SICs, indicating that these responses are indeed mediated by NMDARs. Altogether, these results indicate that neuron–astrocyte communication is enhanced during LFS for LTD induction.

**LTD requires astrocytic $Ca^{2+}$ activity and vesicular release.** We then tested whether astrocytic $Ca^{2+}$ signals modulate LTD of CA3-to-CA1 synapses. To this end, we analyzed the consequences of intracellular infusion of the calcium chelator BAPTA (20 mM) into the astrocytes of the hippocampal slice using a patch pipette. It is well known that BAPTA spreads via gap-junctions into the astrocyte syncytium, interfering with astrocytic $Ca^{2+}$ signaling throughout the slice[9,16] (see Fig. 2a for a representative example of astrocyte intracellular loading with BAPTA and biocytin, followed by streptavidin-Alexa 488 staining, and Supplementary Fig. 1b for widespread quenching of astrocytic $Ca^{2+}$ signals after BAPTA intracellular loading). LTD was then induced with a pairing protocol that combines LFS of SCs and mild postsynaptic depolarization (−40 mV) of the CA1 neuron. This protocol facilitates partial activation of NMDA receptors during synaptic stimulation and does not alter synapses that are not stimulated during depolarization (homosynaptic depression; Supplementary Fig. 3, "control pathway")[17]. This protocol produced robust depression in control slices (Fig. 2b, black graphs). This form of LTD was mediated by NMDARs, as it was blocked by bath-application of AP5 (50 μM) (Fig. 2b, green graphs), but not by the metabotropic glutamate receptor antagonist MCPG (0.5 mM) (Fig. 2b, gray graphs). Strikingly, LTD was strongly impaired in slices where astrocytes were filled with BAPTA (Fig. 2b, red graphs). This impairment was specific for LTD, as LTP induced in CA1 neurons (pairing 3 Hz presynaptic stimulation with 0 mV postsynaptic depolarization) was not altered by interfering with astrocytic $Ca^{2+}$ signaling (Supplementary Fig. 4). This result argues against indirect effects of BAPTA on synaptic plasticity, for example by altering $K^+$ homeostasis or $Ca^{2+}$ channels in neurons. It is also important to keep in mind that BAPTA infusion may not be as effective as a "calcium-clamp" protocol (based on a calcium/EGTA intracellular solution) to block calcium signaling in astrocytes[18]. Nevertheless, under our experimental conditions, intracellular loading with BAPTA was sufficient to significantly reduce LTD without affecting LTP. Lastly, none of these manipulations (AP5, MCPG, BAPTA) altered synaptic transmission in the unpaired (control) pathway (Supplementary Fig. 3).

To further test the idea that NMDAR-dependent LTD requires astrocyte $Ca^{2+}$ elevations, we analyzed slices from inositol-1,4,5-trisphosphate (IP$_3$)-receptor type 2-deficient mice (IP$_3$R2$^{-/-}$)[19], which is the primary IP$_3$R responsible for intracellular $Ca^{2+}$ mobilization in astrocytes[20] (but see also ref. [21]). LTD observed in control conditions was diminished in IP$_3$R2$^{-/-}$ mice (Fig. 2b, blue graphs), and this result was significantly different from the one obtained with control slices ($P < 0.05$, according to Dunn's multiple comparison post-hoc test). This result suggests that NMDAR-dependent LTD induced by low-frequency stimulation requires astrocyte $Ca^{2+}$ signals.

As a complementary approach to test the role of gliotransmission in NMDAR-dependent LTD, we blocked exocytic delivery

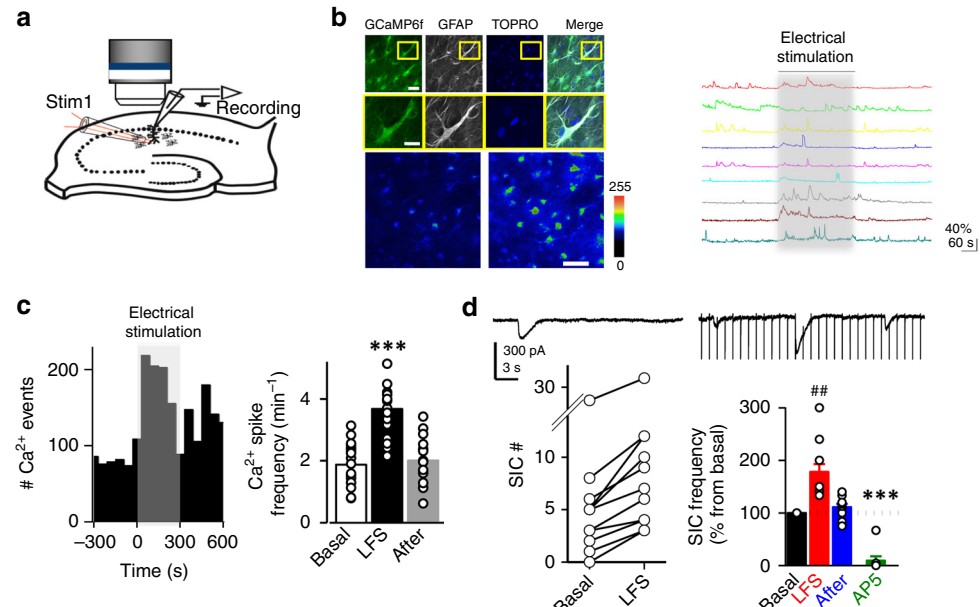

**Fig. 1** Low-frequency stimulation (LFS) induces astrocyte–neuron communication. **a** Scheme showing the experimental configuration for electrical stimulation in *stratum radiatum* and whole-cell recording of CA1 pyramidal neurons. **b** Top, immunocytochemical localization of GCaMP6f (green), GFAP (gray), and TOPRO (blue) in acute hippocampal slices after in vivo injection with AAV5-$P_{GFAP}$-cyto-GCaMP6f. Inset of high magnification of representative astrocyte. Scale bars = 25 and 20 μm, respectively. Representative pseudocolor images (bottom) and intensity $Ca^{2+}$ signals (right) versus time from nine astrocytes, obtained by live imaging from fluorescence intensities of GCaMP6f in hippocampal acute slices before, during, and after LFS protocol (300 pulses at 1 Hz). Scale bar, 50 μm. **c** Left, Representative experiment showing the number of calcium events during an LFS experiment in time bins of 50 s. Right, average spike frequency of $Ca^{2+}$ signals per slice before (baseline), during (LFS) and after LTD induction ($n = 20$) $P < 0.001$ (***). Friedman test followed by Dunn's multiple comparison. **d** Top, representative whole-cell currents from a CA1 pyramidal neuron in an acute hippocampal slice, recorded before (left) and during (right) LFS. (Bottom, left) Number of SICs before and during LFS protocol from 12 different slices. (Bottom, right) summary bargraph of SIC frequency increases in response to LFS. Some slices were treated with the NMDAR antagonist AP5 (green $n = 8$). SICs were compared by Friedman test followed by Dunn's multiple comparison ($^{##}P < 0.01$). AP5 vs. baseline were compared by Mann–Whitney test (***$P < 0.001$)

specifically in astrocytes. Vesicular release of glutamate from astrocytes has been shown to require SNARE protein-dependent mechanisms[22,23]. We then asked whether the inactivation of the astrocytic SNARE machinery interferes with LTD. To inhibit the vesicle-associated SNARE machinery in the astrocyte, we developed an adeno-associated virus (AAV) to express the tetanus toxin light-chain (TeTxLC) and a monomeric Kusabira orange under the GFAP promoter (AAV2/1-$P_{GFAP}$-TeTxLC-2A-mKO; Fig. 2c). Western blot analysis of lysates prepared from infected organotypic slices confirmed a 50% reduction in the levels of VAMP2 and VAMP3 (Supplementary Fig. 5; to note, a full knock-down of these proteins is not expected, since their expression in neurons would not be altered). Expression of TeTxLC in astrocytes of organotypic hippocampal slices virtually abolished SICs in neurons (Fig. 2d), confirming that this approach fully blocks astrocyte-to-neuron communication mediated by glutamate. In addition, we observed that in TeTxLC-infected organotypic slices the LFS protocol failed to produce significant synaptic depression, as observed in control, uninfected, slices (Fig. 2e). Therefore, we conclude that NMDAR-dependent-LTD in the hippocampus requires SNARE protein-dependent vesicular release from astrocytes.

**Optogenetic activation of astrocytes induces LTD**. We next investigated whether the astrocytic activity, without participation of the presynaptic neuron, is sufficient to induce NMDAR-dependent LTD. To explore this idea, the activity of the astrocyte was selectively enhanced using optogenetic tools based on the specific expression of channelrhodopsin-2 (ChR2) in astrocytes[24] by infection of organotypic hippocampal slices with AAVs bearing the

GFAP promoter (AAV5-$P_{GFAP}$-hChR2(H134R)-mCherry, Fig. 3a). Current-clamp recordings from ChR2-expressing astrocytes close to resting membrane potential (−70/−80 mV) showed that astrocytes respond to blue light photostimulation (20 ms pulses) with moderate membrane depolarization (8.2 ± 1.0 mV, $n = 5$ cells; Fig. 3b, inset). Importantly, low-frequency photo-stimulation of ChR2-expressing astrocytes in hippocampal slices (1 Hz during 5 min, in the absence of Schafer collateral stimulation) significantly increased the frequency of intracellular calcium signals in astrocytes (Fig. 3c, left) and their amplitude (Fig. 3c, right).

Next, we investigated whether increasing astrocytic $Ca^{2+}$ activity is sufficient to elicit glutamate-mediated SICs in CA1 neurons from organotypic slices. Selective photo-stimulation of astrocytes induced an increase in the frequency of SICs (Fig. 3d), which was abolished in the presence of the NMDAR antagonist AP5 (50 μM). These results indicate that photo-stimulation of astrocytes elevates their intracellular $Ca^{2+}$ and induces glutamate-mediated gliotransmission leading to NMDAR activation in pyramidal neurons. To test whether this astrocytic activity is sufficient to induce LTD, we recorded CA3–CA1 synaptic responses in organotypic slices before and after low-frequency photostimulation of astrocytes. Importantly, SC stimulation is stopped during the optogenetic induction protocol, and therefore, the activity of the presynaptic neuron is bypassed during induction. To prevent possible forms of synaptic plasticity mediated by activation of presynaptic metabotropic glutamate receptors (mGluRs)[9,25], these experiments were initially carried out in the presence of the broad spectrum mGluR antagonist MCPG (0.5 mM) (but see below experiments without MCPG in Supplementary Fig. 7c). We monitored synaptic responses in CA1 pyramidal neurons evoked by SC stimulation. After baseline

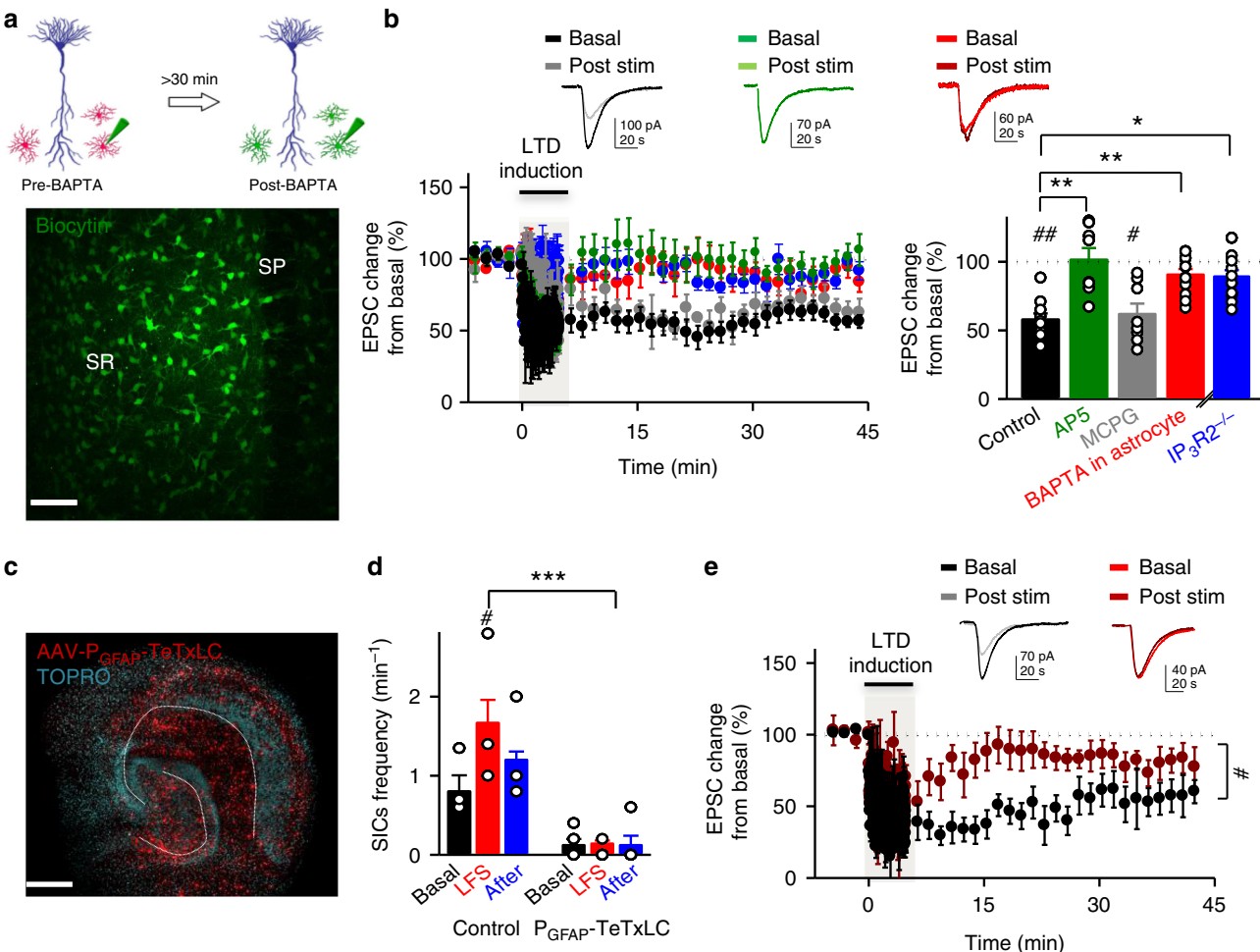

**Fig. 2** NMDA receptor-dependent LTD requires astrocyte activity. **a** Schematic drawing depicting BAPTA dialysis into the astrocytic network from the recorded acute hippocampal slice. BAPTA and biocytin-loaded astrocytes were confocally imaged. Scale bar, 50 μm. **b** (Inset) Representative EPSCs before and 30 min after LFS in acute hippocampal slices. Relative EPSC amplitudes (left) and average relative changes (right) versus time during an LTD experiment with control slices (black, $n = 10$), incubated with AP5 (green, $n = 8$), or MCPG (gray, $n = 8$), with BAPTA-loaded astrocytes (red, $n = 12$) or IP3R2$^{-/-}$ mice (blue, $n = 13$). Wilcoxon tests were used to compare baseline versus 45 min after LFS protocol ($^{\#}P < 0.05$; $^{\#\#}P < 0.01$). Differences between groups were determined by Kruskal–Wallis test followed by Dunn's test ($^{*}P < 0.05$; $^{**}P < 0.01$). Time zero corresponds to the onset of the LTD protocol. **c** Representative fluorescence image showing an organotypic hippocampal slice infected with adeno-associated virus (serotype 2) driving the expression of tetanus toxin light-chain (TeTxLC) fused to the orange fluorescence protein mKO under the GFAP promoter. Scale bar, 200 μm. **d** Summary bargraph of slow-inward current (SIC) frequency in response to LFS in uninfected (Control) organotypic slices ($n = 4$) and in TeTxLC-expressing slices ($n = 5$). Two-way repeated measures ANOVA showed an effect of TeTxLc infection in SICs ($F(1;12) = 51.08$; $P = 0.004$); Bonferroni post-hoc test. SICs in both control and GFAP TeTxLC were also compared by Friedman test followed by Dunn's test. **e** (Inset) Representative EPSCs before and 30 min after LFS in organotypic slices expressing TeTxLC in astrocytes (brown) or in uninfected slices (black). Relative EPSC amplitudes (normalized to baseline values) versus time in control conditions (non-infected; black, $n = 5$) and in slices expressing TeTxLC (dark red, $n = 7$). Time zero corresponds to the onset of the LTD protocol. Mann–Whitney test was used to compare control and GFAP-TeTxLc 45 min after LFS protocol ($^{*}P < 0.05$). For all panels, data are presented as means ± s.e.m. For statistical details and $n$, see Supplementary Table

control recordings, presynaptic stimulation was stopped and photostimulation of ChR2-expressing astrocytes was delivered at 1 Hz during 300 pulses paired with mild postsynaptic depolarization (−40 mV). After photostimulation, when SC stimulation was resumed, we observed a gradual but robust long-lasting depression of synaptic responses (Fig. 4a, red graphs). Light stimulation did not produce significant changes in synaptic transmission in non-infected control slices (Fig. 4a, white graphs), and did not alter neuronal firing (Supplementary Fig. 6a, b) or passive membrane properties in neurons or astrocytes (Supplementary Fig. 6a, c). Therefore, synaptic depression in slices with ChR2-expressing astrocytes was due to selective optogenetic activation of astrocytes, rather than to some nonspecific action of light. Importantly, this astrocyte-driven LTD was blocked by the

NMDAR antagonist AP5 (50 μM) (Fig. 4a, green graphs) and significantly reduced by selectively inhibiting GluN2B-containing NMDARs with ifenprodil (10 μM) (Fig. 4a, purple graphs). Essentially the same results were obtained with acute slices obtained from animals injected in vivo for ChR2-expression in astrocytes (Supplementary Fig. 7a, b). Considered together, these results indicate that low-frequency activation of gliotransmission drives NMDAR-dependent LTD in the hippocampus, similarly to electrically induced LTD.

Photostimulation of astrocytes may drive the release of multiple transmitters, which could produce indirect effects on neuronal NMDARs. In order to rule out most of these effects, we photo-stimulated organotypic hippocampal slices with ChR2-expressing astrocytes while blocking GABAergic, cannabinoid,

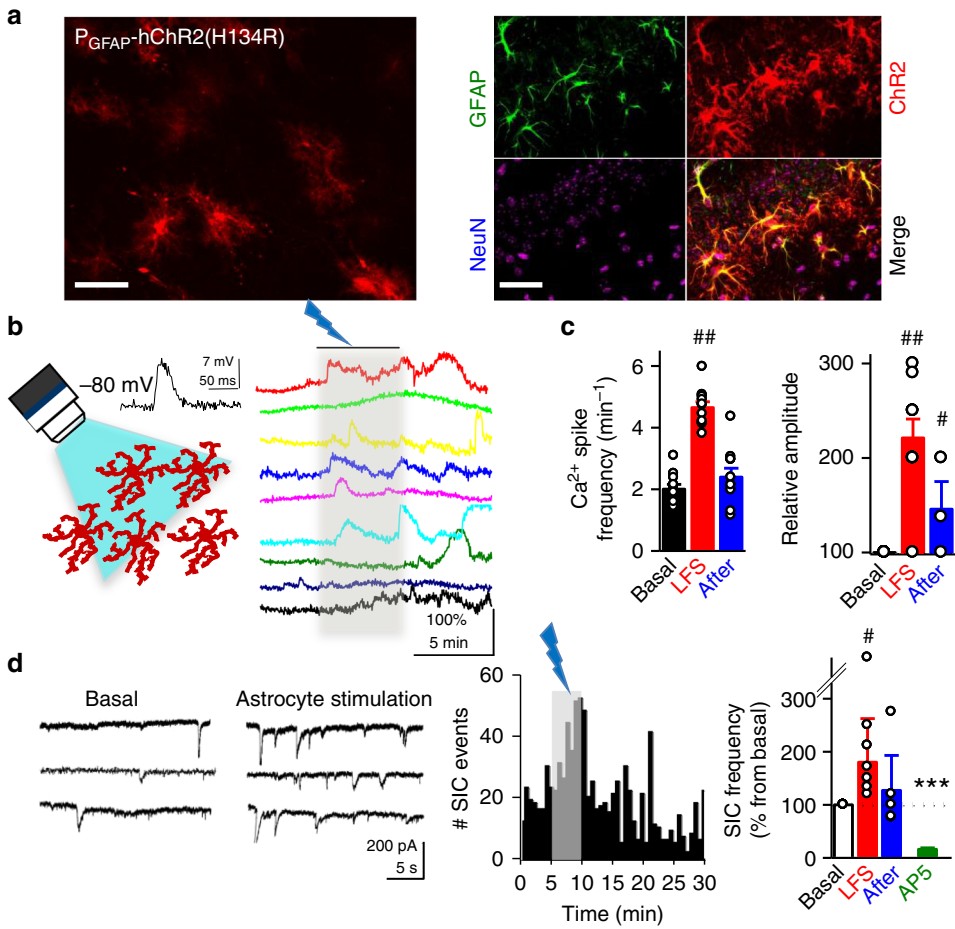

**Fig. 3** Optogenetic stimulation of astrocytes induces astrocyte–neuron communication. **a** (Left) Confocal image of a hippocampal section showing the expression of ChR2-mCherry in astrocytes. (Right) Confocal image of hippocampal organotypic slices immunohistochemically labeled with the astrocytic marker GFAP (green) and the neuronal marker NeuN (cyan). ChR2-mCherry expression is shown in red. Merged image at the lower right corner. Note the selective expression of ChR2-mCherry in GFAP-positive cells (astrocytes). Scale bars, 50 μm (left) and 40 μm (right). **b** (Left) Schematic representation depicting the optical stimulation of ChR2-mCherry-expressing astrocytes, and representative recording of astrocyte membrane potential during optogenetic stimulation (inset) in organotypic hippocampal slices. (Right) Astrocyte $Ca^{2+}$ signals evoked by ChR2 stimulation (1 Hz, 5 min; gray bar). **c** Quantification of the frequency and relative amplitude of astrocyte $Ca^{2+}$ signals before (basal, black bars), during LFS (red bars) and 5 min after (recovery, blue bars) ($n = 69$ astrocytes from 9 organotypic slices). Data were compared by Friedman test followed by Dunn's test ($^{\#\#}P < 0.01$). **d** (Left) Representative example of whole-cell currents from CA1 pyramidal neurons before and during LFS by optogenetic astrocyte stimulation in organotypic hippocampal slices. (Middle) Number of SICs versus time before, during and after optogenetically driven LFS. (Right) Summary of results from experiments as the one shown on the left. Friedman test followed by Dunn's multiple comparison ($^{\#}P < 0.05$). AP5 vs. baseline were compared by Mann Whitney Test ($^{***}P < 0.001$)

cholinergic, purinergic and mGlu receptors with a cocktail of chemical antagonists. Under these conditions, astrocytic photostimulation produced LTD to a similar extent as observed before (Fig. 4b, yellow graphs; compare with Fig. 4a, red graphs). In addition, LTD was also observed in presence of saturating concentrations of D-serine (10 μM) (Fig. 4b, gray graphs), suggesting that D-serine is not the limiting gliotransmitter mediating synaptic depression upon astrocytic photostimulation.

Lastly, we wished to determine how this NMDAR-dependent synaptic depression related with other forms of synaptic modulations driven by astrocytic glutamate. To this end, we carried out the same low-frequency photo-stimulation of hippocampal slices with ChR2-expressing astrocytes, but omitting MCPG from the perfusion solution. As shown in Supplementary Fig. 7c, we still observed a long-lasting depression, but in this case it was preceded by a transient potentiation. These results are consistent with previous reports of facilitation of CA3-to-CA1 synaptic transmission mediated by astrocytic glutamate acting on

presynaptic mGluRs[9,25]. This presynaptic potentiation appears to be then overrun by the NMDAR-dependent depression we are describing here.

**LTD mediated by astrocytes involves postsynaptic mechanisms.** A key aspect of NMDARs during canonical LTD in the hippocampus is their role as a postsynaptic coincidence detector, based on their voltage-dependent blockade by extracellular $Mg^{2+}$. To test whether this property is preserved in astrocytic-driven LTD, we carried out similar photostimulation experiments without postsynaptic depolarization. Under these conditions, LTD was dramatically reduced (Fig. 4b, pink graphs). Furthermore, postsynaptic depolarization was not required for LTD in the absence of extracellular $Mg^{2+}$ (Fig. 4b, black graphs). Taken together, these results indicate that direct stimulation of glutamate-mediated gliotransmission is sufficient to induce NMDAR-dependent LTD in the hippocampus, in which NMDARs act as

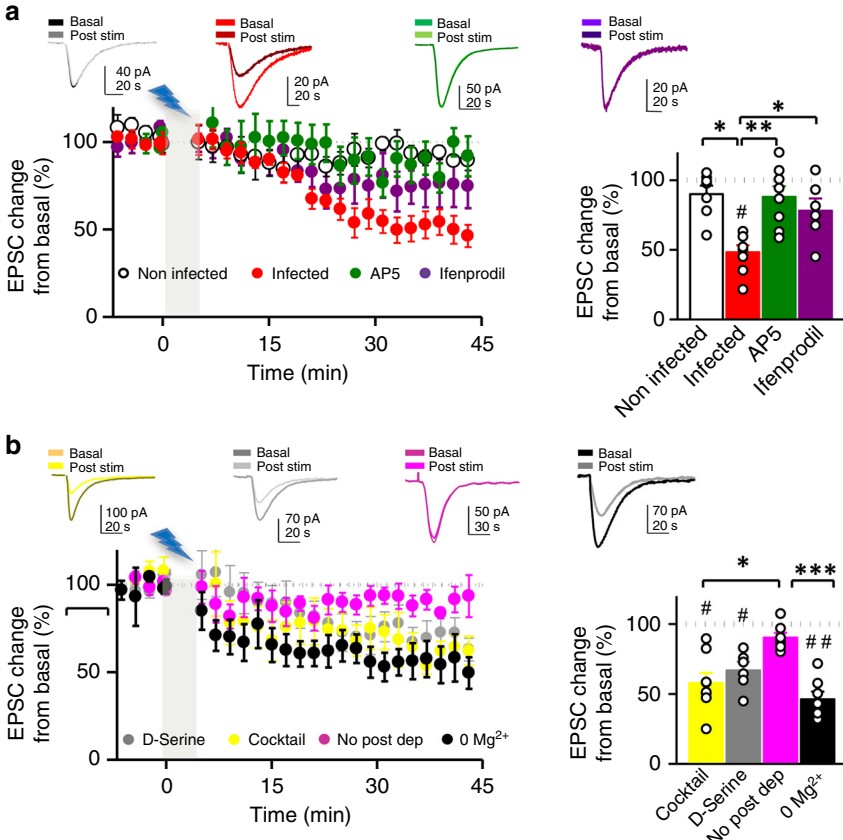

**Fig. 4** Optogenetic low-frequency stimulation of astrocytes induces NMDAR-dependent LTD. **a** (Left) Time course of CA3-CA1 EPSC amplitudes during a photostimulation experiment (gray bar) from control (non-infected) (white, $n = 7$) or Ch2R-expressing organotypic hippocampal slices (red, $n = 8$), in presence of the NMDAR antagonist AP5 (green, $n = 8$) or the GluN2B-selective antagonist ifenprodil (purple, $n = 6$). (Right) Average values of EPSC amplitude relative to baseline, 45 min after the onset of photostimulation. Representative traces with matching colors are shown above the graphs. Wilcoxon was used for assessing LTD expression with respect to baseline ($^\#P < 0.05$). Differences between groups were determined by Kruskal–Wallis test followed by Dunn's test. ($*P < 0.05$; $**P < 0.01$). **b** Relative changes and summary of results of EPSC amplitude over time before and after astrocyte photostimulation in the presence of saturating concentrations of D-serine (10 μM, gray, $n = 6$); a cocktail containing GABA$_A$ (picrotoxin, 50 μM), GABA$_B$ (saclofen, 100 μM), P2 purinergic (suramin, 100 μM), muscarinic cholinergic (atropine, 50 μM), CB1 cannabinoid (AM251, 2 μM), A1 adenosine (CPT, 10 μM), and metabotropic glutamate (MCPG, 0.5 mM) receptor antagonists (yellow, $n = 8$); without postsynaptic depolarization (pink, $n = 7$) and without postsynaptic depolarization in 0 Mg$^{2+}$ (black, $n = 7$), in organotypic hippocampal slices. Differences between groups were determined by Kruskal–Wallis test followed by Dunn's test ($*P < 0.05$, $***P < 0.001$). Wilcoxon statistical test was used to analyze LTD expression with respect to baseline ($^\#P < 0.05$, $^{\#\#}P < 0.05$). Representative traces with matching colors are shown above the graphs. For all panels, data are presented as mean ± s.e.m

coincidence detectors of glutamate release and postsynaptic depolarization.

To further test that postsynaptic NMDARs are mediating astrocyte-induced LFS LTD, we introduced the use-dependent NMDAR channel blocker dizocilpine (MK-801) specifically into the postsynaptic neuron of acute hippocampal slices using the patch pipette[10]. We then performed paired recordings from two adjacent pyramidal neurons (Fig. 5a). One recording pipette included 1 mM MK-801, whereas the other contained standard intracellular solution, as control. LTD was induced by either electrical SC stimulation (Fig. 5a, left panels) or astrocyte photostimulation (Fig. 5a, right panels). As expected, synaptic depression was clearly observable in control neurons (Fig. 5a, b, green graphs) for both electrical and photo-stimulation. In contrast, under our experimental conditions, MK-801-infused neurons failed to express any LTD (Fig. 5a, b, blue graphs). These data strongly support the interpretation that NMDARs located in the postsynaptic neuron are required for astrocyte induced-LTD (but see current debate on the metabotropic functions of NMDARs and the ability of MK-801 to block LTD[26]).

The expression of NMDAR-mediated LTD in the hippocampus involves the endocytic removal of AMPARs from the postsynaptic membrane[3,27]. To test whether this process is also elicited upon astrocytic stimulation, we examined changes in AMPAR distribution before and after photostimulation (1 Hz, 300 pulses) of astrocytes (this experiment was carried out without extracellular Mg$^{2+}$ to allow NMDAR activation without patching the postsynaptic neuron for depolarization). To monitor AMPAR distribution, we overexpressed EGFP-GluA2 in organotypic hippocampal slices and analyzed receptor accumulation in spines and dendrites based on EGFP fluorescence intensity. In addition, surface receptor expression was assessed by anti-GFP immunostaining under non-permeabilizing conditions[28] (Fig. 5c; see Supplementary Fig. 8 for selectivity of the surface immunostaining). We found that the spine/dendrite ratios for both EGFP fluorescence intensity (total receptor) and anti-GFP labeling (surface expression) were decreased upon astrocyte stimulation (Fig. 5d), confirming that astrocyte-driven LTD results in persistent removal of AMPARs from spines and the postsynaptic membrane.

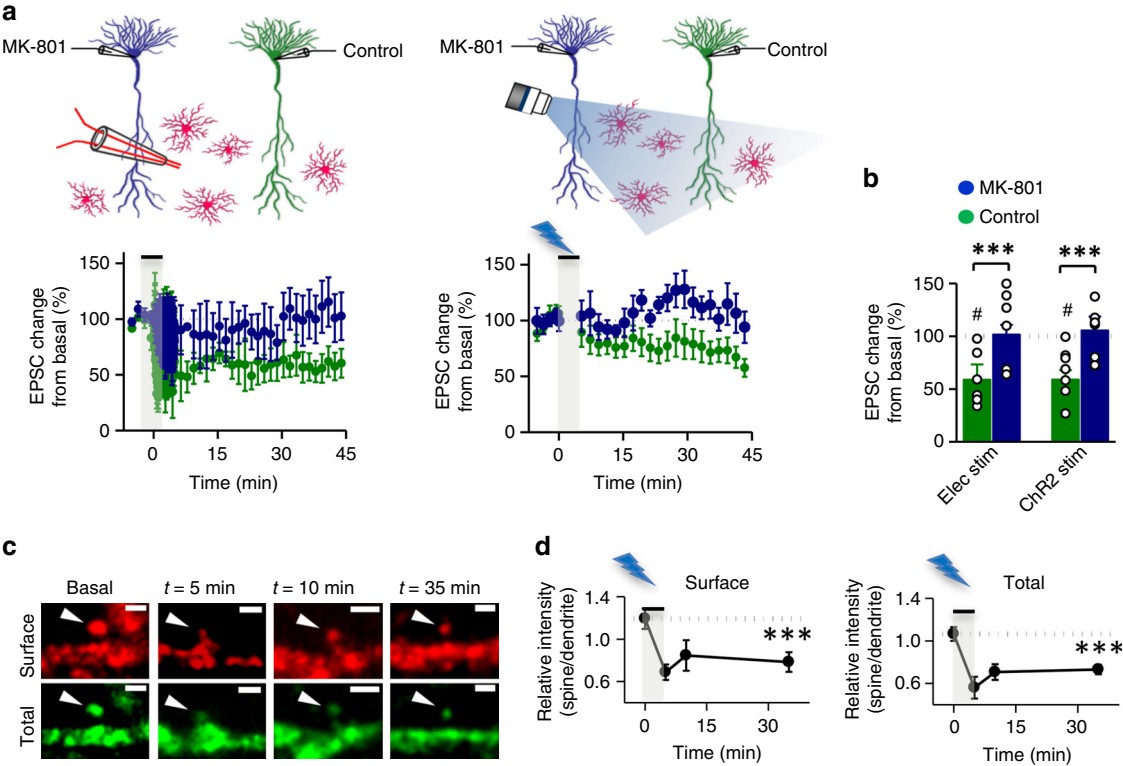

**Fig. 5** NMDAR-dependent LTD induced by astrocytes is expressed postsynaptically. **a** (Top) Schematic representation showing the experimental design for LFS with electric stimulation (left) and ChR2-astrocyte photostimulation (right), from control (green) or MK-801-filled (1 mM, blue) neurons, in acute hippocampal slices. (Bottom) Relative EPSC changes from basal values *versus* time from the experiments represented on top ($n = 6$ for electric stimulation, and n = 7 for ChR2-astrocyte stimulation). **b** Summary of results from the average change in EPSC 45 min after the onset of LFS protocol. No differences were observed by two-way ANOVA between electric and ChR2-astrocyte stimulation. A significant effect was found for MK-801 $F(1, 22) = 14.64$; $P = 0.0009$. Wilcoxon statistical test was used to analyze LTD expression with respect to baseline ([#]$P < 0.05$). **c** Representative confocal images of dendritic spines from CA1 neurons expressing EGFP-GluA2 in organotypic hippocampal slices. Green channel represents total receptor expression (EGFP signal) and red signal represents surface expression (anti-GFP immunostaining under non-permeabilizing conditions). Images were acquired before (basal) and at different times after photostimulation. Scale bars, 1 μm. **d** Quantification of spine/dendrite ratios for surface (left panel) or total (right panel) EGFP-GluA2 expression from neurons before and after photostimulation ($n_{t=0} = 365$; $n_{t=5} = 197$; $n_{t=10} = 393$; $n_{t=35} = 116$ spines from $n = 3$ experiments). Time zero corresponds to the onset of the photostimulation protocol. Differences between groups were determined by Kruskal–Wallis test followed by Dunn's test ([***]$P < 0.001$). Data are presented as means ± s.e.m

**Astrocytic p38α MAPK is necessary for LTD at CA3–CA1 synapses.** To further ascertain the role of astrocytes in LTD, we evaluated their participation in the signaling mechanisms that have been shown to be involved in LTD. p38 MAPK is a well-known integrator of cellular signaling[29]. It is also considered a major factor for stress signaling in the brain in response to multiple extracellular stimuli[30]. Interestingly, p38 MAPK has been implicated in synaptic depression, both NMDAR-dependent and mGluR-dependent[31–33], as well as in synaptic depression associated to Alzheimer's disease[34–36]. However, it has not been possible to pinpoint the precise role of p38 MAPK in LTD, or even its potential place of action (see for example ref. [37]). In addition, the involvement of p38 MAPK has been typically tested with pharmacological inhibitors that do not distinguish between the p38α and p38β family members[38], both expressed in neurons and glial cells. To address this issue, we have employed acute hippocampal slices from isoform-specific knockout mice (Fig. 6a–c). Specific deletion of p38β (p38β$^{-/-}$)[39] did not affect basal synaptic transmission (Fig. 6d) nor LTD expression induced by LFS (Fig. 6e), as compared to wild-type littermates (compare black and green graphs). Therefore, these results rule out p38β as a mediator of LTD. We then used a double knockout combining the total p38β deficiency (p38β$^{-/-}$) with a conditional deletion of p38α under the control of the neuronal $Ca^{2+}$/calmodulin-

dependent protein kinase II alpha (CaMKIIα) promoter (p38α (neuronal)$^{-/-}$)[35]. Basal transmission was normal in these animals (Fig. 6d, red symbols). Surprisingly, LTD was still present, and even slightly enhanced as compared to wild-type littermates (Fig. 6e, red graphs). These results strongly suggest that neither p38β nor neuronal p38α is required for NMDAR-dependent LTD in the hippocampus.

The experiments described above were carried out with a neuronal-specific deletion of p38α because the constitutive knockout is embryonic lethal[40]. Therefore, these results leave open the intriguing possibility that astrocytic p38α could mediate hippocampal NMDAR-dependent LTD. To test this hypothesis, we deleted p38α in a cell type-selective manner using mice with a floxed p38α allele (p38α$^{lox/lox}$)[41] and adeno-associated viruses (serotype 5) encoding the Cre recombinase under the astrocytic GFAP promoter (AAV5-P$_{GFAP}$-GFP-Cre) or the neuronal CaMKIIα promoter (AAV5-P$_{CaMKII}$α-mCherry-Cre). First, we confirmed the specificity of these viruses for Cre expression in astrocytes and neurons, respectively, by infecting organotypic hippocampal slice cultures (Fig. 7a and Supplementary Fig. 9). We then used this system to determine the contribution of astrocytic and neuronal p38α to conventional, electrically induced NMDAR-dependent LTD. Strikingly, LTD was virtually absent in p38α$^{lox/lox}$ organotypic slices expressing Cre in the astrocytes

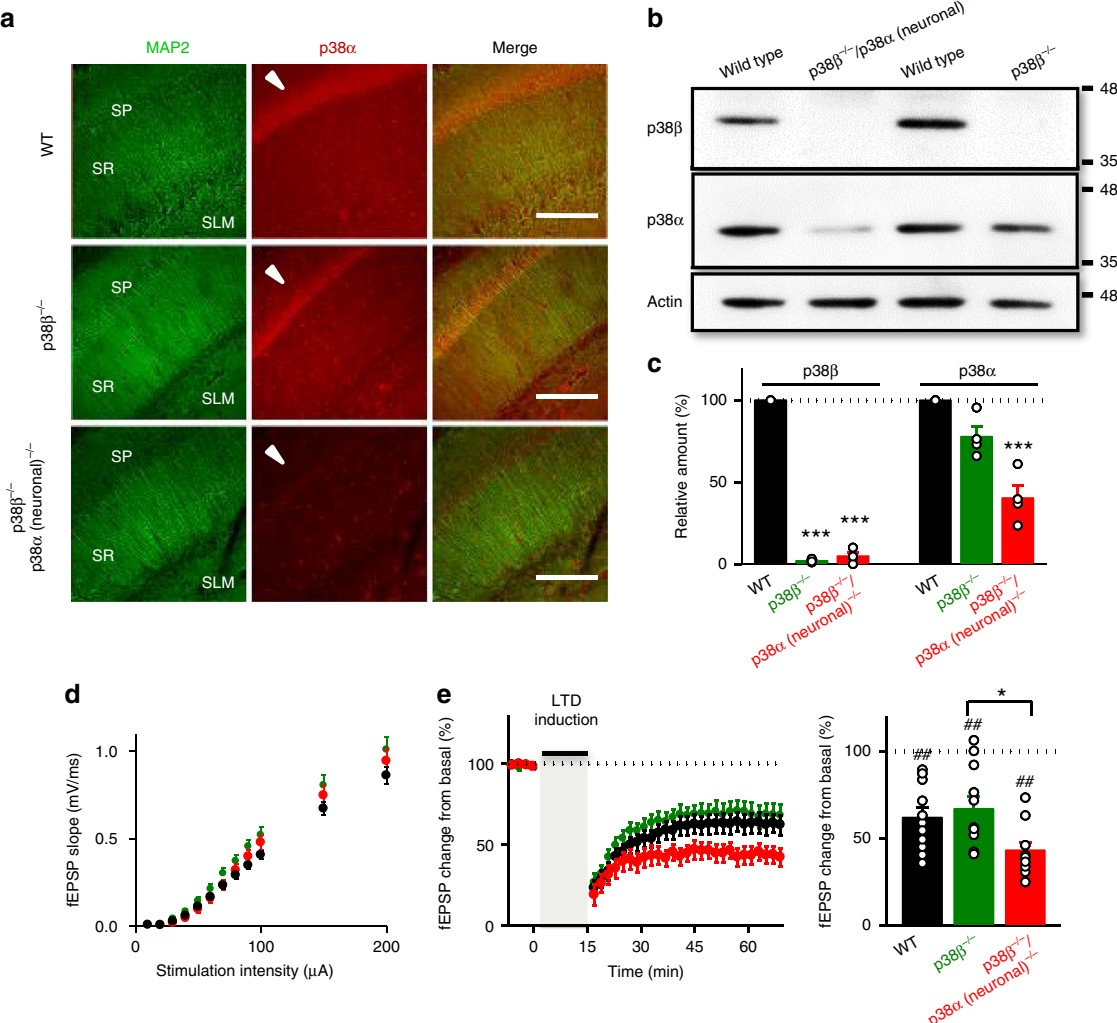

**Fig. 6** Neuronal p38α and p38β MAPK are not necessary for NMDAR-dependent LTD. **a** Immunohistochemistry of hippocampal sections from WT, p38β$^{-/-}$, and p38β$^{-/-}$/p38α (neuronal)$^{-/-}$ animals with anti-MAP2 antibody (green) and anti-p38α antibody (red). SP: *stratum pyramidale* (indicated with an arrowhead in middle panels). SR: *stratum radiatum*. SLM: *stratum lacunosum moleculare*. Scale bars, 200 μm. **b** Western blot analysis of p38α and p38β protein levels from WT, p38β$^{-/-}$, and p38β$^{-/-}$/p38α (neuronal)$^{-/-}$ hippocampus. Actin served as a loading control. **c** Relative amount in percentage of p38β and p38α from WT (black bars, n = 6), p38β$^{-/-}$(green bars, n = 4), and p38β$^{-/-}$/ p38α (neuronal)$^{-/-}$ (red bars, n = 4) from western blots as the one shown in **b**. Kruskal–Wallis test followed by Dunn's test (***P < 0.001). **d** Input–output curves presenting fEPSP slope from acute hippocampal slices in response to increasing stimulus strength (n > 39 slices from n > 7 mice). No differences were found by two-way ANOVA (F(2, 240); P = 0.73). **e** (Left) Time course of relative changes in fEPSP slope before and after LFS in acute hippocampal slices from WT (black, n = 11 from 7 mice), p38β$^{-/-}$/p38α neuronal)$^{-/-}$ (red, n = 9 slices from 7 mice) and p38β$^{-/-}$ (green, n = 11 from 7 mice) knockout mice. (Right) Summary of results from the end of the time courses shown on the left. Differences between groups were determined by Kruskal–Wallis test followed by Dunn's test (*P < 0.05). Wilcoxon statistical test was used to analyze LTD expression with respect to baseline (##P < 0.01)

(AAV5-$P_{GFAP}$-GFP-Cre; Fig. 7b, green graphs). In contrast, LTD was preserved in slices expressing Cre in neurons ($P_{CaMKIIα}$-mCherry-Cre; Fig. 7b, red graphs). To note, the blockade of LTD with GFAP-driven Cre cannot be explained by non-specific expression of Cre in neurons with the GFAP promoter, because full neuronal expression of Cre with the CaMKIIα promoter did not have any effect on LTD. Importantly, this LTD was still p38-dependent, as it was blocked with the p38α/p38β inhibitor SB203580 (5 μM) in both neuronal Cre-expressing slices (Fig. 7b, dark blue graphs), and in uninfected slices (Fig. 7b, light blue graphs).

Taken together, these results unambiguously demonstrate that astrocytic, but not neuronal, p38α MAPK is required for LTD in the hippocampus.

**p38α MAPK is required for the increase in astrocytic glutamate release after LFS**. To further understand the role of p38α in LTD, we tested whether astrocytic p38α was responsible for astrocyte-to-neuron communication mediated by glutamate during LFS. To this end, we used the fluorescence glutamate sensor iGluSnFR[42] specifically expressed in astrocytes with an AAV bearing the GFAP promoter (AAV5-$P_{GFAP}$-iGluSnFR) (Fig. 8a). Using this sensor, we were able to detect spontaneous glutamate signals on the surface of astrocytes (Fig. 8b, c). These events developed over time scales of seconds (Fig. 8b and Supplementary Fig. 10), which is at least one order of magnitude slower than those previously reported for synaptically released glutamate from neurons[42]. Therefore, we are confident that these events represent slow glutamate release from astrocytes.

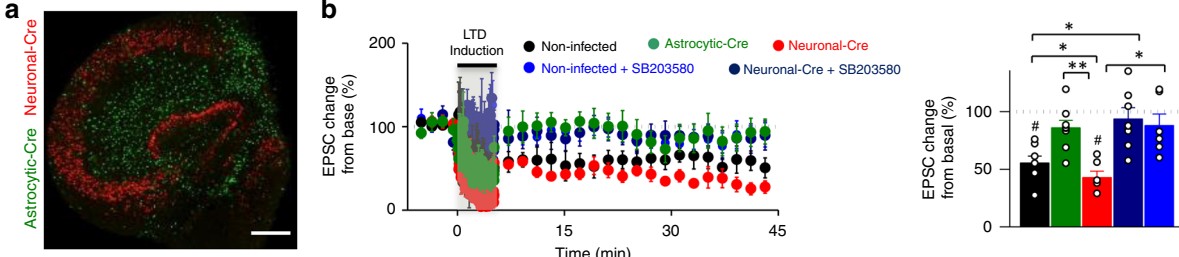

**Fig. 7** Astrocytic p38α MAPK is necessary for NMDAR-dependent LTD. **a** Representative fluorescence image of an organotypic hippocampal slice from a p38α$^{lox/lox}$ mouse simultaneously infected with neuronal AAV5-P$_{CaMKIIα}$-mCherry-Cre (Neuronal-Cre) and astrocytic AAV5-P$_{GFAP}$-GFP-Cre (Astrocytic-Cre). Scale bar, 200 μm. **b** Relative EPSC amplitudes (normalized to baseline values) versus time and average relative changes of EPSC amplitudes 45 min after electrical-LTD induction in control p38α$^{lox/lox}$ organotypic slices (non-infected; black, $n = 7$), in the presence of the p38α/p38β inhibitor SB203580 (non-infected + SB203580; light blue, $n = 7$), or in slices infected with astrocytic-Cre (green, $n = 8$), with neuronal-Cre (red, $n = 6$), or with neuronal-Cre in the presence of SB203580 (Neuronal-Cre + SB203580; dark blue, $n = 6$). Differences between groups were determined by Kruskal–Wallis test followed by Dunn's test, (*$P < 0.05$). Wilcoxon statistical test was used to analyze LTD expression with respect to baseline ($^{\#}P < 0.05$). Data are presented as means ± s.e.m. For statistical details and $n$, see Supplementary Table

Control experiments were carried out after expressing Cre in astrocytes from p38α$^{lox/+}$ organotypic hippocampal slices (and therefore retaining one p38α allele). The frequency of astrocytic iGluSnFR spikes was significantly increased during electrical LFS (see representative traces in Fig. 8c, analysis and quantification in Fig. 8d, e), without significant changes in spike amplitude or duration (Supplementary Fig. 10). Notably, the increase in glutamate spikes during LFS was abolished when Cre was expressed in organotypic slices from p38α$^{lox/lox}$ animals (and therefore devoid of astrocytic p38α) (Fig. 8d, e). This failure to increase glutamate release was also reflected in astrocyte-to-neuron communication. Thus, the increase in the frequency of SICs during LFS was also absent in organotypic slices lacking astrocytic p38α, as compared to slices retaining p38α (Fig. 8f, g).

We also evaluated the requirement of p38 for astrocyte-to-neuron communication under standard conditions of LFS-induced LTD (in the presence of Mg$^{2+}$ and without glycine). To this end, we carried out current-clamp recordings in acute hippocampal slices and monitored neuronal membrane potential during LFS. For these experiments, we initially brought the resting membrane potential of the neuron to −40 mV with small current injections to mimic the mild depolarization used for LTD induction under voltage-clamp configuration. Under these conditions, LFS elicited small EPSPs timed with the presynaptic stimulation, as expected (Supplementary Fig. 11a, inset). In addition, and superimposed with these synaptic responses, we observed large and slow depolarizations that occurred sporadically (representative trace, labeled with asterisks, in Supplementary Fig. 11a), with an average frequency of 1.5 min$^{-1}$. Given their kinetics and frequency, we consider these responses to be the membrane potential correlates of the SICs. Importantly, these responses were also dependent on NMDARs and p38 activity, as tested with their respective inhibitors (quantification shown in Supplementary Fig. 11b).

**p38α deletion from astrocytes enhances long-term memory.** Finally, we evaluated whether selective deletion of astrocytic p38α had cognitive consequences in vivo. It has been previously reported that molecular manipulations that impair LTD prevent the decay of long-term memories[43,44]. Conversely, experimental LTD induction inactivates previously acquired memories[45]. Therefore, we tested whether deletion of p38α in astrocytes or in neurons had an effect on long-term memory. To this end, we bilaterally injected AAVs expressing Cre either in neurons (CaMKIIα promoter) or in astrocytes (GFAP promoter) into the hippocampus of p38α$^{lox/lox}$ mice (Fig. 9a). Four weeks after infection, we observed Cre-expressing cells, which were distributed along the entire hippocampus. The rate of infection, as evaluated by colocalization with astrocytic (GFAP) or neuronal (NeuN) markers (Supplementary Fig. 12a–c) was 48.6 ± 6.6% ($N = 7$) for astrocytic Cre, and 89 ± 5.2% ($N = 5$) for neuronal Cre (Supplementary Fig. 12d, e). Confirming our previous results in organotypic slice cultures, LTD was impaired in acute slices after in vivo deletion of p38α in hippocampal astrocytes (Supplementary Fig. 13, green graphs). In contrast, LTD was preserved in both vehicle-injected (Supplementary Fig. 13, black graphs) and neuronal Cre-expressing mice (Supplementary Fig. 13, red graphs).

We then assessed hippocampal-dependent long-term memory in p38α-depleted animals using contextual fear conditioning. First, uninjected p38α floxed mice were placed in a novel context, and after 150 s of familiarization, they were exposed to five electrical foot shocks (Fig. 9b, "d0"). 48 h later (Fig. 9b, "d2"), mice were tested for freezing behavior (pre-infection test) and then allocated to one of three experimental conditions to be injected the following day with vehicle ($N = 11$), neuronal Cre virus ($N = 9$) or astrocytic Cre virus ($N = 11$) (Fig. 9b, "d3"). Animal allocation was designed so that the three groups were matched for freezing behavior during training session (Supplementary Fig. 14a) and at the day before injection (Supplementary Fig. 14b; pre-infection test, "d2"). In addition, mice that were trained similarly but not shocked during the conditioning displayed very low levels of freezing response (d2, pre-infection test: 0.9 ± 0.1%, $N = 2$; d30, post-infection test: 0.8 ± 0.1%) indicating that freezing observed in conditioned mice reflects a conditioned fear response.

Thirty days after injection, mice were returned to the chamber and contextual fear memory was measured (post-infection test; Fig. 9b, "d30"). Notably, we found that specific deletion of p38α in astrocytes significantly increased the freezing response compared to that observed in vehicle-injected and neuronal Cre-expressing mice (Fig. 9c). In addition, we also compared the freezing response for each individual mouse before infection (d2) and 30 days after infection (d30). For both, vehicle-injected and neuronal Cre-expressing mice, the rate of freezing had not significantly changed at 30 days, although there was a trend towards lower freezing in both groups. In contrast, mice with astrocytic removal of p38α displayed a significant increase in freezing response at d30 as compared to d2 (Fig. 9d). Importantly, none of the groups were altered in general levels of activity, grooming or spontaneous freezing (Supplementary Fig. 14c–f),

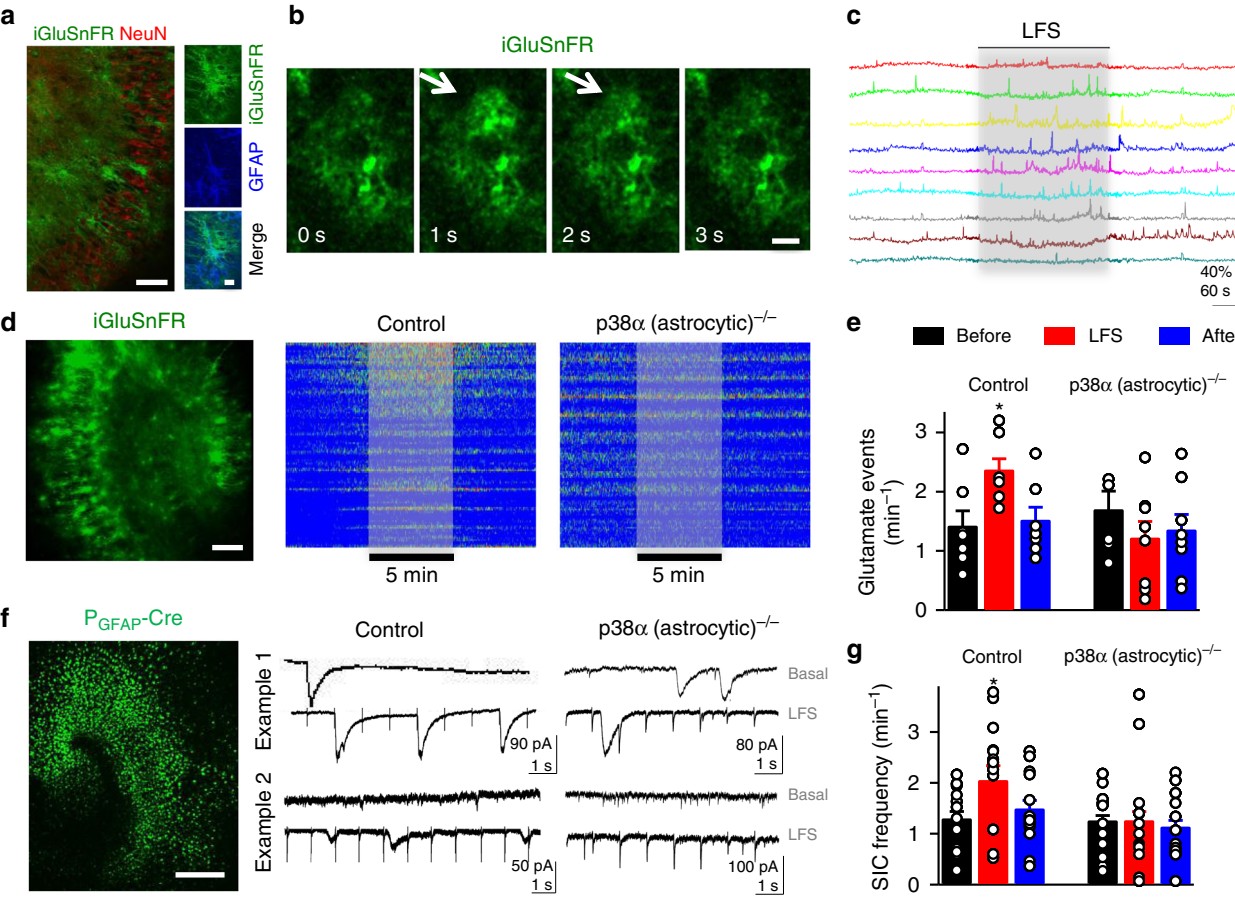

**Fig. 8** Astrocytic p38α is necessary for LFS-induced astrocyte-to-neuron communication. **a** Representative immunohistochemical localization of iGluSnFR, NeuN, and GFAP in organotypic hippocampal slices infected with AAV5-$P_{GFAP}$-iGluSnFR. Right panels shown at higher magnification. Scale bars, 50 μm (left) and 30 μm (right). **b** Representative two-photon images of iGluSnFR fluorescence from individual astrocytes throughout time taken at 1 Hz. Arrows indicate a transient and local increase in glutamate-induced fluorescence. Scale bar, 20 μm. **c** Representative glutamate signals from the astrocytic surface before, during (gray beam) and after electrical LFS. Note the increased number of astrocyte iGluSnFR spikes during LFS. **d** Left, representative fluorescence image showing an organotypic hippocampal slice expressing iGluSnFR under GFAP promoter. Scale bar, 300 μm. Right, representative kymograph of glutamate activity (ROIs pooled from two different experiments were used), color-coded according to fluorescence changes from p38α$^{lox/+}$ slices (left panel, Control) or from p38α$^{lox/lox}$ slices (right panel, p38α(astrocytic)$^{-/-}$) infected with $P_{GFAP}$-Cre virus. The period of LFS is highlighted. **e** Quantification of frequency of glutamate events from images as the ones represented in **d**. Glutamate events in control and GFAP-Cre infected p38α$^{lox/lox}$ slices were compared by Friedman test followed by Dunn's multiple comparison (*$P < 0.05$). Two-way repeated measures ANOVA demonstrated a significant interaction between AAV infection and the time of glutamate events. $F(2;39) = 3.43$, $P = 0.04$. Bonferroni post-hoc test: control vs. p38α (astrocytic)$^{-/-}$. Data are presented as means ± s.e.m. **f** Left, Confocal image of an organotypic hippocampal slice from p38α$^{lox/lox}$ animals after 12 days viral transfection with AAV5-$P_{GFAP}$-GFP-Cre. Scale bar, 300 μm. Right, two examples of whole-cell currents from pyramidal neurons in p38α$^{lox/+}$ (left, Control) or p38α$^{lox/lox}$ (right, p38α(astrocytic)$^{-/-}$) organotypic slices infected with $P_{GFAP}$-Cre. **g** Frequency of SICs in basal conditions (before, black), during LFS (red) and after LFS (after, blue), binning of 5 min, quantified from recordings as the ones shown in **f**. SICs before, during and after LFS were compared by Friedman test followed by Dunns post-hoc test. Two way repeated measures ANOVA demonstrated a significant effect for AAV infection in SICs $F(1;87) = 5.32$; *$P < 0.05$. Bonferroni post-hoc test. Data are presented as means ± s.e.m. For statistical details see Supplementary Table

nor in anxiety-related behavior (as assayed in an open field; Supplementary Fig. 15), in the absence of previous fear conditioning. These control experiments strongly suggest that the increase in freezing response in astrocytic p38α-deleted animals is reflecting an enhancement of long-term memory of fear conditioning.

Finally, to strengthen the causal relationship between astrocytic p38α removal and cognitive performance, we examined the freezing response of individual mice according to the fraction of infected astrocytes in their hippocampi (measured as shown in Supplementary Fig. 12). As shown in Fig. 9e, a positive correlation was detected between the percentage of infected astrocytes (GFP/GFAP positive cells) and the freezing response 30 days after infection. Taken together, these results strongly suggest that reduced levels of astrocytic p38α cause an

enhancement in long-term memory, which is consistent with the role of astrocytic p38α as a mediator of synaptic plasticity and cognitive function.

## Discussion
The data in this study challenges the standard view that NMDAR-dependent LTD in the hippocampus is due to the direct communication between the presynaptically released glutamate and the postsynaptic activation of NMDARs. Instead, we now provide conclusive evidence that astrocytes are obligatory inter-mediates for LTD expression in hippocampal CA1, which relies on astrocytic p38α MAPK signaling.

The observation that glutamate released from astrocytes can activate postsynaptic NMDARs is not new, as this is the original

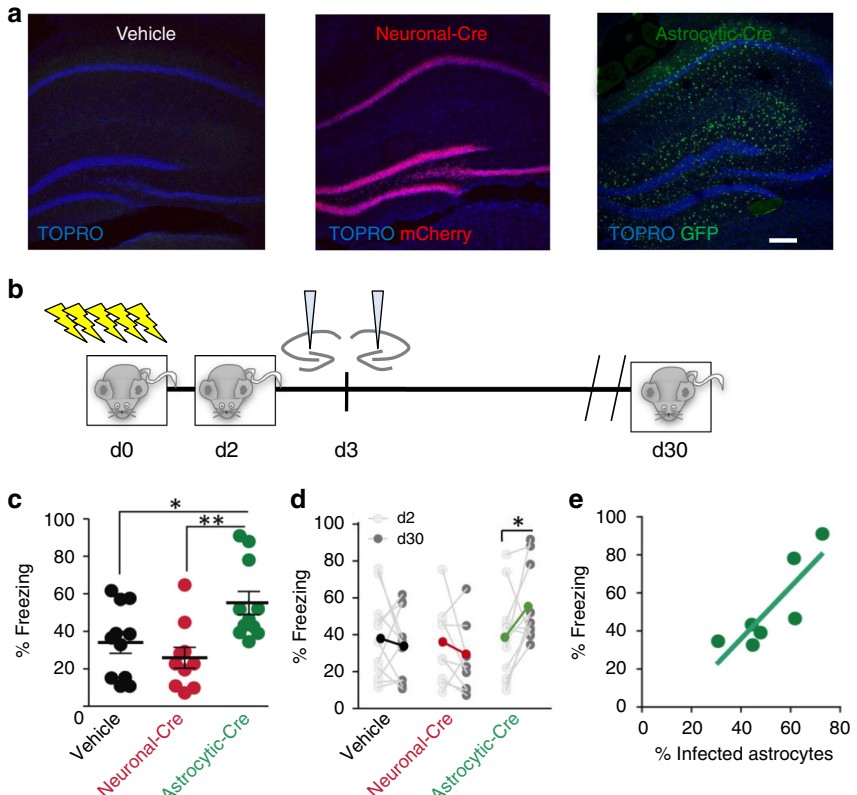

**Fig. 9** Deletion of astrocytic p38α enhances long-term memory. **a** Representative fluorescence image of hippocampal sections from p38α$^{lox/lox}$ mice 4 weeks after in vivo delivery of AAV5-P$_{CaMKIIα}$-mCherry-Cre (Neuronal-Cre) and astrocytic AAV5-P$_{GFAP}$-GFP-Cre. Scale bar, 150 μm. Counterstaining with *TOPRO* for cell nuclei was used to visualize hippocampus morphology. **b** Cartoon representing the contextual fear-conditioning paradigm used in this study. p38α$^{lox/lox}$ mice were subjected to contextual fear conditioning on the training day (d0) consisting of five foot shocks. Two days later, mice were briefly tested (3 min re-exposure) for fear retrieval (pre-infection test; d2) to check freezing levels before infection. The next day (d3), mice were allocated in one of the three experimental groups: saline or either AAV-expressing Cre under CaMKIIα (neuronal Cre) or under GFAP promoter (astrocytic-Cre) was bilaterally delivered in both dorsal and ventral hippocampus. 30 days after conditioning, infected mice were tested for freezing behavior (post-infection test, d30). **c** Percentage of freezing observed 30 days after contextual fear conditioning in p38α$^{lox/lox}$ control mice (vehicle-treated) and p38α$^{lox/lox}$ infected with neuronal-Cre (red symbols) and astrocytic-Cre virus (green symbols). Differences were determined by Kruskal–Wallis test followed by Dunn's test (*$P < 0.05$, **$P < 0.01$). **d** Comparison of freezing levels displayed by each animal and group at the pre-infection (d2) and post-infection retrieval tests (d30). Two-way ANOVA demonstrated a significant interaction between the test and the infection ($F(2; 28) = 0.57$, $P = 0.009$). Bonferroni post-hoc test (*$P < 0.05$). **e** Spearman correlation between the percentage of infected astrocytes in the *stratum radiatum* of CA1 (GFAP-positive cells) and the percentage of freezing displayed by p38α$^{lox/lox}$ mice infected with astrocytic-Cre virus during post-infection test ($r = 0.785$; $P = 0.048$). Linear regression is also displayed in the graph

interpretation of astrocytic-derived SICs[46]. This phenomenon has also been shown to produce synaptic depression when triggered by exogenous cannabinoids acting on astroglial CB$_1$ receptors[11]. Furthermore, spike-timing dependent plasticity has also been shown to be mediated by cannabinoid-induced release of glutamate from astrocytes, in this case acting on presynaptic NMDARs[10]. Nevertheless, what we are proposing here is a reinterpretation of the classical mechanism for CA3-to-CA1 hippocampal LTD induced by LFS and mediated by postsynaptic NMDARs. We now believe that astrocytic glutamate is an intrinsic driver of this mechanism. This conclusion stems from the striking observation that interfering with astrocytic Ca$^{2+}$ signaling strongly reduces LTD. To note, this manipulation did not affect NMDAR-dependent LTP of CA3–CA1 synapses. This result is in contrast with a previous report on the importance of Ca$^{2+}$-dependent D-Serine release from astrocytes for LTP expression[18]. Although we cannot be sure of the reason for this discrepancy, it is worth mentioning that LTP expression is less dependent on D-Serine when NMDAR activation is facilitated by postsynaptic depolarization[47], as it is the case of our patch-clamp protocol for LTP induction. In addition to astrocytic Ca$^{2+}$

signaling, we also concluded that hippocampal LTD requires SNARE-dependent vesicular release from astrocytes, as it is blocked by TeTxLC.

The field of astrocyte-to-neuron communication is riddled with controversies, mostly because of the scarcity of precise methods to interfere with specific astrocytic functions[6,7,48–51]. To avoid some of these caveats, we have proceeded to selectively delete a candidate gene in either astrocytes or neurons[52]. We show that genetic ablation of p38α MAPK virtually abolishes LTD when targeting astrocytes, while producing no effect (or slightly enhancing LTD) when targeting neurons. p38α is an important regulator of multiple cellular responses, which often depend on the cell type[29]. Our results indicate that astrocytic p38α is essential for activity-dependent glutamate release from astrocytes and for astrocyte-to-neuron communication. Previous work has implicated p38α MAPK signaling in synaptic plasticity[53], but the underlying mechanism had not been resolved. Particularly, the cell type in which p38 signaling operates was uncertain, as bath-application of the p38α/p38β inhibitor SB203580 blocked LTD[32,33], whereas intracellular infusion of the same inhibitor into the postsynaptic neuron had no effect[37]. Our results now clarify

this apparent contradiction, by concluding that p38α activity specifically within the astrocytic compartment is required for LTD expression in the postsynaptic neuron.

It is also intriguing how the relatively sporadic release of glutamate from the astrocyte (SIC frequency of 1–2 min$^{-1}$) is able to elicit LTD, while fast repetitive release from the presynaptic neuron (1 Hz during LFS) is not. On this point, it is worth considering that SICs give rise to much larger charge transfers than regular EPSCs. In other words, NMDARs on the postsynaptic neuron may sense as much glutamate from astrocytic release as from the presynaptic neuron (when relative frequencies and charge transfers are taken into account). Obviously, these considerations do not rule out a joint contribution to LTD induced by LFS. For example, we have noticed that LFS in the absence of astrocytic release leads to a fast but transient synaptic depression. In contrast, exclusive activation of astrocytes produces a gradual but long-lasting depression. Therefore, it is tempting to speculate that these two cellular components (neuronal and astrocytic) contribute to different phases of synaptic plasticity induced by LFS. Namely, the fast-paced release of presynaptic glutamate may be responsible for the initial synaptic depression, and the slower release of glutamate from the astrocyte would then ensure the persistence of this depression.

According to this scenario, another arising question is how glutamate released from astrocytes achieves synapse-specific plasticity, so that only synapses activated during LTD induction undergo depression (homosynaptic depression[1]). On the one hand, we should keep in mind that the relative contribution of synaptic and extrasynaptic NMDARs for LTD is not clear yet[54]. In addition, the anatomical details of gliotransmitter release should be considered. Particularly, vesicular structures in astrocytes have been observed forming clusters within processes, in close proximity to synapses[55,56] and in the vicinity of neuronal NMDARs[16], supporting the possibility of single-synapse precision in astrocytic regulation. In any case, even though much structural information is still missing, from a functional point of view, it has already been shown that astrocytes can regulate neuronal function with exquisite synapse and circuit selectivity in the central amygdala[57] and in basal ganglia[58].

Finally, we show that removal of p38α from hippocampal astrocytes has an enhancing effect on long-term memory in vivo. In fact, it was surprising to find that the fear response in these animals increases over time. It is possible that p38α contributes to cognitive function by multiple mechanisms. Nevertheless, it is reasonable to speculate that this increase in long-term memory is related with the requirement of astrocytic p38α for LTD. An increase in fear memory over time has been previously reported as a consequence of memory generalization[59–61]. Interestingly, this process is generally limited by new learning and memory flexibility[62], which are thought to rely on LTD[2,63–66]. Therefore, even though the complexities of astrocytic signaling and synaptic plasticity are still being unraveled, we speculate that deletion of astrocytic p38α produces an enhanced expression of fear memory over time because of the impairment in LTD mechanisms.

In conclusion, we have presented evidence for a novel scenario in NMDAR-dependent synaptic plasticity in the hippocampus, where astrocytic signaling is revealed as an obligatory relay between the presynaptic and postsynaptic neuron for changes in synaptic strength related to cognitive function.

## Methods

**Mice.** All the biosafety procedures for handling and sacrificing animals were approved by the bioethics committee from the Consejo Superior de Investigaciones Científicas (CSIC), and followed the European Commission guidelines for the welfare of experimental animals (2010/63/EU, 86/609/EEC). Animals of both genders were used, and were housed in standard laboratory cages with ad libitum access to food and water, under a 12:12 h dark–light cycle in temperature-controlled rooms. C57BL/6; p38α$^{lox/lox}$; p38$^{-/-}$; p38β$^{-/-}$ and p38α (CaMKIIα)$^{-/-}$); IP3R2$^{-/-}$ mice and Wistar rats were used. All behavioral assays were carried out in a blind manner, so that the experimenter and the person carrying out the analyses did not know the genotype or the treatment that the animal had received. Data analysis was carried out separately for males and females. The results were similar and they have been pooled together.

**Slice preparation.** Acute hippocampal slices were obtained from mice (13–18 days or 2–3 months old, both sexes, for whole-cell and field recording experiments, respectively). Animals were anaesthetized and decapitated. The brain was rapidly removed and placed in ice-cold artificial cerebrospinal fluid (ACSF). Slices (350 μm thick) were incubated during >1 h at room temperature (21–24 °C) in ACSF containing (in mM): NaCl 119, KCl 2.5, NaH$_2$PO$_4$ 1, MgCl$_2$ 1.2, NaHCO$_3$ 26, CaCl$_2$ 2.5, and glucose 11, and was gassed with 95% O$_2$/5% CO$_2$ (pH = 7.3).

Organotypic hippocampal slice cultures were prepared from Wistar rats or p38α$^{lox/lox}$ mice (postnatal day 5–6, both sexes) and placed on porous membranes in a medium containing 1 mM L-glutamine, 1 mM CaCl$_2$, 2 mM MgSO$_4$, 5.2 mM NaHCO$_3$, 30 mM Hepes, 13 mM glucose, and 1 mg l$^{-1}$ insulin, 0.0012% ascorbic acid, 20% horse serum, and maintained at 35.5 °C under 5% CO$_2$. Culture medium was replaced with a fresh one every 2–3 days. The slices were used at 9–16 days in vitro. Slices were infected in the CA1 area through pulse injection of the adeno-associated viruses (serotype 5) with the help of a picospritzer at 1 day in vitro and maintained for 9–16 days in culture. The slices were recorded in ACSF containing (in mM): NaCl 119, KCl 2.5, NaH$_2$PO$_4$ 1, glucose 11, NaHCO$_3$ 26, MgCl$_2$ 4, and CaCl$_2$ 4, gassed with 95% O$_2$/5% CO$_2$ (pH = 7.3).

**Electrophysiology.** Unless otherwise indicated, ACSF was supplemented with 0.05 mM picrotoxin and 0.5 mM MCPG to block GABA$_A$ and mGluRs, respectively. Recordings from CA1 pyramidal neurons were made, for most experiments, using the whole-cell patch-clamp technique. Patch electrodes had resistances of 4–10 MΩ when filled with the internal solution that contained (in mM) for pyramidal neurons: KGluconate 135, KCl 10, HEPES 10, MgCl$_2$ 1, ATP-Na$_2$ 2 (pH = 7.3). In some experiments astrocytes were patched with 4–9 MΩ electrodes filled with an intracellular solution containing (in mM): 1 MgCl$_2$, 8 NaCl, 2 ATP, 0.4 GTP, 10 HEPES, 20 BAPTA titrated with KOH to pH 7.2–7.25 and adjusted to 275–285 mOsm. The Ca$^{2+}$ chelator BAPTA was included to prevent Ca$^{2+}$ elevations in astrocytes. Astrocyte whole-cell recordings lasted at least 30 min to allow the dialysis of BAPTA throughout the gap-junction connected astrocyte network. Recordings were obtained with Multiclamp 700 A/B amplifiers and pClamp software (Molecular Devices). Electrophysiological properties were monitored before and at the end of the experiments. Series and input resistances were monitored throughout the experiment using a −1 mV pulse. Recordings were considered stable when the series and input resistances, and stimulus artifact duration did not change > 20%. Cells that did not meet these criteria were discarded. Cells were visualized under an Olympus BX50WI microscope (Olympus Optical, Tokyo, Japan) with a ×60 water immersion objective.

For recording SICs, the extracellular Mg$^{2+}$ was equimolarly substituted by Ca$^{2+}$ and 10 μM glycine was added to optimize NMDAR activation. SICs were analyzed using the Event Detection protocols in the Clampfit routine of pClamp (the template consisted of a 5 ms baseline, 22 ms rise time, 100 ms decay time). The average kinetics of these events was: rise time 27.4 ± 4.8 ms and decay time 152.3 ± 16.0 ms (n = 23).

Synaptic responses were evoked with bipolar electrodes using single-voltage pulses (200 μs, up to 20 V). Stimulating electrodes were theta capillaries of (4–9 μm tip) filled with ACSF, or platinum/iridium cluster electrodes of 25 μm in diameter. The stimulating electrodes were placed over SC fibers between 150 and 250 μm from the recorded cells. For whole-cell recordings, stimulation strength was set to elicit EPSCs of 50–100 pA in size. NMDAR-dependent LTD was induced using a pairing protocol by stimulating SC fibers at 1 Hz (300 pulses) while depolarizing the postsynaptic cell to −40 mV unless indicated otherwise. LTP was induced using a pairing protocol by stimulating SC fibers at 3 Hz (300 pulses) while depolarizing the postsynaptic cell to 0 mV.

Field excitatory postsynaptic potentials (fEPSPs) were recorded with glass electrodes filled with ACSF (0.2–0.8 MΩ) placed in CA1 *stratum radiatum*. fEPSPs were recorded at different stimulation intensities for each slice to generate an input–output curve. This curve was also used to set the baseline fEPSP value at ≈70% of the maximum for LTD experiments. LTD was induced at 1 Hz (900 pulses).

Synaptic plasticity changes were determined as the normalized change in average response size during the last 4.5 min of recording (41.5–45 min after induction protocol) compared to 6 min baseline. To illustrate the time course synaptic parameters were grouped in 1.5-min bins.

**Optogenetic stimulation.** 9–16 days after viral injection, cells expressing mCherry-ChR2 were readily visualized from their red fluorescence, in a confocal laser scanning mode at 543 nm with a HeNe laser. For imaging experiments, blue-light stimulation (20 ms) was delivered by an external laser at 1 Hz during 5 min (300 light pulses) using diode-pumped solid-state blue laser with analog intensity

control (473 nm, 200 mW, MBL-III-473, OptoEngine, LLC) coupled via SMA terminal to a 200-µm fiber (ThorLabs) that was placed over the slices for full-field stimulation. For electrophysiology experiments, blue-light pulses (20 ms at 1 Hz during 5 min) were delivered with the mercury lamp of the microscope, using the GFP filter set.

**Ca²⁺ imaging**. $Ca^{2+}$ levels in astrocytes located in the hippocampus were monitored with a multiphoton laser scanning microscope (AxioImager M, LSM510, Zeiss Oberkochen, Germany) equipped with a pulsed infrared laser set to 780 nm (Spectra Physics Mai-Tai, Prairie Technologies, USA) using $Ca^{2+}$ indicator fluo-4 (Molecular Probes, Eugene, OR) or microinjection of AAV5-$P_{GFAP}$-cyto-GCaMP6f; PENN Vector Core; viral titer $6.13 \times 10^{13}$). GCaM6f was transfected in organotypic hippocampal astrocytes or in vivo for imaging of acute hippocampal slices. After 2 weeks of viral injection, specific expression of constructs in the astrocytes was confirmed by immunostaining. Some slices were incubated with fluo-4-AM (2–5 µl of 2 mM dye were dropped over the hippocampus, attaining a final concentration of 2–10 µM and 0.01% of pluronic acid) for 15–20 min at room temperature. mCherry signal was acquired in a confocal laser scanning mode at 543 nm with a HeNe laser. Astrocytes showing both mCherry and GCaMP6f positive labeling were selected for analysis, and regions of interest (ROI) were selected from the GCaMP6f image. Images were acquired every 0.5–1 s at 30–32 °C. $Ca^{2+}$ variations were estimated as changes of the fluorescence signal over baseline ($\Delta F/F_0$), and regions of interest were considered to respond to the stimulation when $\Delta F/F_0$ increased three times the standard deviation of the baseline. The astrocyte $Ca^{2+}$ signal was quantified from the $Ca^{2+}$ oscillation frequency. The time of occurrence was considered at the onset of the $Ca^{2+}$ spike. The $Ca^{2+}$ signal frequency was obtained from the number of $Ca^{2+}$ spikes occurring in 6–14 astrocytes in the field of view during periods before (basal) and after LTD stimulation. To test the effects of ChR2 stimulation on $Ca^{2+}$ spike signal under different conditions, the respective mean basal (5 min before ChR2 stimulation) and maximum $Ca^{2+}$ spike signal from different slices were averaged and compared. Mean values were obtained from at least four slices in each condition.

**Glutamate imaging**. Imaging of iGluSnFR-expressing astrocytes was performed in hippocampal slices with a multi-photon (AxioImager M, LSM510, Zeiss Oberkochen, Germany) laser scanning system (tuned to 870 nm) equipped with a Spectra Physics MaiTai (Prairie Technologies, USA) and with epifluorescence system using a CCD camera attached to the microscope. Cells were illuminated using CoolLED pE-100 fluorescent excitation system and images were acquired every 0.75–1.5 s. Images were acquired at 1 Hz frame rate for 15 min. Minor drift in the *XY* plane of image stacks was post hoc corrected using TurboReg (ImageJ plugin). For kymograph analysis, ROIs were designed using the Grid plugin of ImageJ (44 × 44 px, equivalent to 14–25 × 14–25 µm). The values for these ROIs are then analyzed using the GECIquant open source plugin for Fiji, to detect and analyze glutamate signals in astrocytes expressing genetically encoded iGluSnFr.

**Plasmid construct and TeTxLC virus purification**. We generated a recombinant AAV[67] bearing a GFAP promoter[68] and a synthetic codon-optimized tetanus toxin light chain (TeTxLC)[69]. For expression analyses of targeted cells by fluorescence imaging, the TeTxLC sequence was followed with a 2A peptide sequence and a monomeric Kusabira orange (mKO)[70]. The 2A peptide bridge sequence derived from Thosea asigna virus[71] allows for expression of multiple genes under the same promoter[72]. The final plasmid, pAAV-2/1$P_{GFAP}$-TeTxLC-2A-mKO, was co-transfected with pDp1 (for serotype 1), pDp2 (for serotype 2) helper plasmids in HEK293 cells. 72 h after transfection, HEK293 cells were collected and packaged viruses were released by repeated freeze (−80 °C) and thaw cycles as described previously[73]. Viruses were purified by pre-casted 1-ml Heparin columns (Amersham, Freiburg, Germany); serotype 2 capsid protein binds to heparin, enabling AAV purification. Quality of purified AAVs were determined by SDS–PAGE that shows three bands corresponding to the viral capsid proteins with expected molecular weights of 87 kDa (VP1), 73 kDa (VP2), and 62 kDa (VP3). Subsequently, functional virus titers were determined in primary neuron cultures (~2.3 × $10^8$ infectious units per ml).

**Western blot**. Hippocampal extracts for Western blot analysis were prepared in lysis buffer containing 10 mM HEPES pH 7.4, 150 mM NaCl, 10 mM EDTA, 1% Triton, phosphatase inhibitors cocktail (Roche), and a protease inhibitor cocktail (Roche). Samples were homogenized at 4 °C, and protein content was determined by the Pierce® BCA Protein Assay kit (Thermo Scientific). Total protein (20–40 µg) was processed on 8% SDS–PAGE and transferred to a PVDF membrane (Immo-blot-P Millipore). After blocking with 5% non-fat dried milk, blots were incubated at 4 °C overnight with primary antibodies anti-p38α MAPK (Santacruz, sc-535), anti-p38β MAPK (Thermo Scientific, 33-8700), anti-actin (Millipore, MAB1501R), anti-VAMP3 Thermo Scientific, PA1-767A), and anti-synaptobrevin (Synaptic Systems, clone 91-1). Corresponding secondary antibodies (peroxidase-conjugated anti-rabbit or anti-mouse; Jackson ImmunoResearch 711-035-152, Sigma I5381) were incubated for 1 h. Detection was carried out by chemiluminescence (Immo-bilon Western, Millipore) using ImageQuant™ LAS 4000 mini biomolecular imager.

**In vivo virus delivery and confirmation of virus expression**. AAV5-CamKIIα-mCherry Cre and AAV5-$P_{GFAP}$-GFP-Cre (adenovirus serotype 5, $4.8 \times 10^{12}$ viral particles per ml; Gene Therapy Vector Core at University of North Carolina) were used. Two months old p38α$^{lox/lox}$ mice were anesthetized with isoflurane 2% in oxygen and placed in a custom adapted stereotaxic apparatus. Stereotaxic bilateral injections (600 nl; 30 nl/min) were made in both dorsal (AP: 1.80 mm ML: ± 1.3 mm DV: 1.5 mm) and ventral hippocampus (AP: −2.8 mm ML: ±2.4 mm DV: 1.9 mm). After injection, the pipette was held in place for 5 min prior to retraction to prevent leakage, and then removed and skin sutured. The animal was allowed to recover from anesthesia with the help of heating pads and was returned to the cage once it showed regular breathing and locomotion. 1 month after viral injection, the location of the virus was confirmed on the basis of mCherry and GFP expression and specific cell type expression of constructs was confirmed by immunostaining.

**Contextual fear conditioning**. Training and testing took place in a rodent observation cage (30 × 37 × 25 cm) composed of three walls of stainless steel, and a plexiglas door. The floor was composed of 20 steel rods of 4 mm of diameter 1.5 cm equidistant from each other. The cage was placed in a sound-attenuating box with ventilation fans that provided a constant background noise of 68 dB and constant illumination (20 W). Contextual fear conditioning was performed 72 h before in vivo infection. During conditioning, mice were placed in the chamber, and after 150 s of familiarization, they received five scrambled foot shocks of 0.75 mA of intensity from a shock generator LI100-26 Shocker (LETICA I.C.) (2 s duration, 1 min apart). Mice were removed from the chamber 1 min after the last shock. 48 h after conditioning, animals were tested during 3 min (pre-infection test, d2) and then allocated to different groups for infection 24 h later. 1 month after conditioning, mice were placed in the chamber for 12 min and memory was tested again (post-infection test). Behavior was recorded by overhead cameras and memory was assessed by the duration of freezing behavior. The freezing response is defined as the absence of movement in the mouse except for the movements required for breathing. Data was expressed as the % of freezing response during the first 3 min of test.

**Tissue preparation for immunofluorescence**. Mice were perfused transcardially with phosphate buffer (0.1 M) followed by 4% paraformaldehyde (PFA). Brains were post-fixed in 4% PFA and transferred to 30% sucrose. Brains were embedded in 4% of agarose and coronal sections (70 µm) were cut using a vibratome (Leica VT1200S). A 1/6 section-sampling fraction was used to create six sets (each containing sections at 410 µm intervals) for immunohistochemistry.

**Immunohistochemistry**. The slices were fixed with 4% PFA, 4% sucrose in PBS for 2 h at room temperature. Non-specific binding was blocked with 3% horse serum, 3% BSA, 0.1% Triton, for 2 h in PBS. Triton was omitted for immunostaining under non-permeabilizing conditions. Samples were then incubated with the corresponding primary antibodies for 2 days and secondary antibodies for 1 h at RT. The primary antibodies used were: anti-p38α (Santa Cruz, sc-535), anti-MAP2 (Covance, PCK-554P), anti-GFAP (Sigma, G3893), anti-Neu-N (Abcam, ab177487), anti-Iba1 (Wako, 019-19741), and anti-GFP (Roche, 11814460001). Secondary antibodies were anti-rabbit Alexa 647 (Thermo Fisher, A-31573), anti-chicken Alexa 488 (Thermo Fisher, A-11039), anti-mouse Alexa 488 (Thermo Fisher, A-21121), and anti-mouse Alexa 555 (Thermo Fisher, A-21137). Samples for imaging were mounted with Prolong Diamond Antifade (Thermo Fisher, P-36970).

**Image acquisition, processing, and quantification**. Fluorescence images were acquired with a Zeiss ZM510 confocal microscope using Zeiss Zen software. Image analysis was carried out with Image J software (public domain software developed at the US National Institutes of Health).

**Quantification of EGFP-GluA2 spine/dendrite distribution**. Circular ROIs were generated around spine heads and the adjacent dendritic shafts. Integrated fluorescence intensity was quantified in each compartment, corresponding to the spine and the dendrite, after background subtraction. Spine/dendrite ratios were calculated from these values. This method is internally normalized for immunostaining variability, since immunofluorescence values are always acquired in pairs of spine and adjacent dendrite. Additionally, we always selected spine–dendrite pairs from the GFP channel, avoiding any bias with respect to their surface immunostaining.

**Quantification of Cre expression**. Cells positive for Cre expression were detected from their mCherry or GFP fluorescence, respectively. The cellular specificity of Cre expression under GFAP and CaMKIIα promoter was tested by immunohistochemical analysis of mCherry and GFP cells with the neuronal maker NeuN and the astrocytic marker GFAP. Colocalization was confirmed by orthogonal projection of z-stack files. The percentage of infection along rostro-caudal hippocampal axis was estimated by confocal z-stack images at ×40 lens acquired from *stratum pryramidale* and *stratum radiatum* of CA1 (from six serial sections per animal (70 µm; 420 µm apart; from bregma −1.00 mm to bregma −3.30 mm)). mCherry-Cre-positive and GFP-Cre-positive cells and the number of infected neurons and

astrocytes were counted manually in frames of 226.7 µm × 226.7 µm (1024 × 1024). Data are expressed as the percentage of infected neurons or astrocytes or infected neurons and astrocytes per 1000 µm².

**Statistical analysis**. Data are expressed as mean ± SEM. Statistical analysis of the LTD expression with respect to baseline transmission was performed using a Wilcoxon's tests $P < 0.05$ (#), $P < 0.01$ (# #), and $P < 0.001$ (# # #). Differences between groups were determined by non-parametric Mann–Whitney $U$-test, One-way ANOVA (non-parametric Kruskall–Wallis) followed by Dunn's test or two-way ANOVA followed by Bonferroni post hoc test ($P < 0.05$ (*), $P < 0.01$ (**), and $P < 0.001$ (***)). For repeated measures, Friedman test or two-way repeated measures ANOVA were used. Correlation analysis was performed by non-parametric Spearman's test. Statistical differences were performed using Sigmaplot 11.00 and/or GraphPad Prism 7. Statistical details for each quantitative experiment are illustrated in Supplementary Table.

**Drugs, chemicals, and AAVs**. 1,2-bis(2-aminophenoxy)ethane-N,N,N′,N′-tetra-acetate (BAPTA), (RS)-α-Methyl-4-carboxyphenylglycine (MCPG), N-(Piperidin-1-yl)-5-(4-iodophenyl)-1-(2,4-dichlorophenyl)-4-methyl-1H-pyrazole-3-carboxamide (AM251), and SB203580 were purchased from Tocris Cookson and Fluo-4-AM from Molecular Probes (Eugene, OR). AAV5-$P_{GFAP}$-hChR2(H134R)-mCherry; AAV5-$P_{GFAP}$-GFP-Cre, AAV5-$P_{CaMKIIα}$-mCherry-Cre from UNC Gene Therapy Center Vector Core and AAV5-$P_{GFAP}$-iGluSnFR and AAV5-$P_{GFAP}$-cyto-GCaMP6f from Penn Vector Core. All other drugs were from Sigma.

**Reporting summary**. Further information on research design is available in the Nature Research Reporting Summary linked to this article.

## Data availability

The data that support the findings of this study are available from the corresponding author upon reasonable request.

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

## Acknowledgements

We thank the personnel at the fluorescence microscopy facility of the CBMSO (SMOC) for their expert technical assistance. We also thank Alfonso Araque, Carlos Dotti, Liset Menéndez de la Prida, Manuel Valero, Sara Mederos and Gertrudis Perea for expert advice and critical reading of the manuscript, and Simon Arthur (University of Dundee, UK) for the p38 knockout mice. We thank Godwin K. Dogbevia, Artur Luzgin, and Maria Calleja for technical help with molecular biology, virus purifications and characterization of AAV2/1-P$_{GFAP}$-TeTxLC-2A-mKO. This work was supported by grants from the Spanish Ministry of Economy and Competitiveness to J.A.E. (SAF2015-72988-EXP, PCIN-2016-095 and SAF2017-86983-R), to M.N. (SAF2014-58598-JIN; RYC-2016-20414), to M.I.C. (IJCI-2015-25507), and to J.A.E. and A.R.N. (CSD2010-0045). M.N. was also funded from BBVA Foundation and L'Oreal Unesco "For Women in Science".

## Author contributions

M.N. carried out most of the experimental work. M.I.C. carried out the behavioral experiments. R.P. carried out the electrophysiology recordings on p38β and conditional p38α knockouts. S.C. and A.R.N. provided these animals and advised on p38 analysis. J.E.D. carried out the AMPA receptor imaging experiment, A.K. carried out immuno-histochemical and Western blot analysis of GFAP-Cre and CaMKIIα-Cre infected slices. I.S. assisted with some of the in vivo viral injections and electrophysiological recordings. S.C.-C. and M.T.H. generated AAV2/1-P$_{GFAP}$-TeTxLC-2A-mKO virus and carried out its initial characterization. M.N., M.I.C., R.P., and J.E.D. analyzed data. M.N., M.I.C., and J.A.E. designed research and wrote the paper.

## Additional information

**Competing interests:** The authors declare no competing interests.

**Peer Review Information:** *Nature Communications* thanks H. Rheinallt Parri, and the other, anonymous, reviewer(s) for their contribution to the peer review of this work. Peer reviewer reports are available.

