## [Peer Review File · Nature Communications]

Reviewers' comments:

Reviewer #1 (Remarks to the Author):

This manuscript presents experiments designed to test the role of astrocytic vesicular glutamate release in low frequency stimulation induced hippocampal LTD, and associated memory processing. The authors use an impressive array of electrophysiological, multiphoton fluorescence imaging, molecular manipulations and behavioral studies to test the necessity and sufficiency for hippocampal LTD of stimulation-induced astrocytic Ca²⁺ spiking and resultant vesicular glutamate release. The authors report the following:

- 1) In cultured hippocampal slices transfected with GCaMP6f under the GFP promoter, local low frequency (1Hz) electrical stimulation in the region of the Schaeffer collaterals increases the frequency of detectable Ca²⁺ spikes.
- 2) Similar stimulation also elicited NMDA receptor-dependent slow inward currents (SICs) in patch-clamped CA1 pyramidal cells (all done in zero Mg²⁺ and added glycine).
- 3) Local injection of BAPTA (20mM) into a single astrocyte, which resulted in presumed GAP-junction mediated spread to other astrocytes (evidenced by spread of fluorescent dye also included in the patch injector) reduced/prevented subsequent induction of LTD in a patch-clamped (-40mV) CA1 pyramid, as did bath application of the NMDA receptor antagonist, AP5, but not the mGluR antagonist MCPG.
- 4) Such astrocyte BAPTA loading did not prevent induction of LTP, induced by 3Hz stimulation and stronger depolarization of the CA1 pyramid (0mV).
- 5) None of the above manipulations affected transmission amplitude in an unpaired stimulated pathway.
- 6) Induction of LTD was "diminished" in IP3-receptor knockout mice.
- 7) Transfection of cultured hippocampal slices with GFAP driven tetanus toxin to prevent vesicular release reduced the frequency of spontaneous and stimulus-evoked SICs, and reduced/alterd the time course of LTD recorded in CA1 pyramids.
- 8) Selective light stimulation of astrocytes, via photo-activation of channelrhodopsin transfected under the GFAP promoter depolarized astrocytes, and when repeated at low frequency increased Ca²⁺ spiking in astrocytes, SICs in CA1 pyramidal cells, and induced LTD in CA1 pyramidal cells (when clamped at -40mV during photostimulation), with the latter two events being blocked by AP5. Such LTD was also reduced, but not blocked by the NR2B selective antagonist Ifenprodil.
- 9) Astrocyte photostimulation-induced LTD was not prevented by a cocktail of antagonists designed to block GABAergic, cannabinoid, cholinergic, purinergic and mGlu receptors, or by the addition of D-serine at concentrations expected to fully activate the NMDAR.
- 10) Astrocyte photostimulation-induced LTD did not occur when the cell was not depolarized to -40mV, unless extracellular Mg²⁺ was removed, and it was blocked by postsynaptic MK-801 (applied in the patch pipette, which also blocked electrical stimulation induced LTD).
- 11) Astrocyte photostimulation-induced a translocation of EGFP-tagged GluA2 subunits of AMPAR from

spines to dendrites as expressed by the ratio of EGFP fluorescence in the two compartments, as well as by subsequent surface staining by antibodies under non-permeabilizing conditions.

12) The authors manipulated cultured slices with a variety of molecular tools to delete P38 (a or b), known to be crucial for LTD induction, either globally or selectively in neurons or astrocytes, and only astrocyte selective P38a or broad spectrum antagonism significantly attenuated LTD.

13) In vivo injections of viruses into various cre-lines that selectively diminished astrocytic p38a, but not selective ablation of neuronal p38a prevented subsequent induction of LTD in acutely prepared hippocampal slices, and enhanced the magnitude of freezing responses after 30 days after shock-induced conditioned fear, an effect that correlated with efficacy of astrocytic p38a knockdown.

Based on these observations, the authors conclude that “the action of astrocytic glutamate on postsynaptic NMDARs is not only a superimposed effect of astrocytes on excitatory synapses, with important physiological consequences of its own, but that this is the intrinsic mechanisms by which classical (low frequency-induced) NMDAR-dependent LTD operates.”

Such a conclusion would obviously be revolutionary to our thinking about memory storage and the broad field of neuron-astrocyte signaling. Accordingly, the topic and results of this manuscript will be of widespread interest to the neuroscience community and readership of Nature Communications. Moreover, as touched on above, the authors have used an impressive array of tools to systematically test most of the proposed sequential mechanisms, all of which generally support their provocative conclusion.

However, as the authors acknowledge, this is currently a highly contentious issue generally, and to draw such radical conclusions requires exemplary experimental support. In this context, there are several substantive issues that lessen my enthusiasm for this manuscript, some of which can be directly experimentally tested, and thus should be. In particular, while everything the authors tested shows correlation, necessity and sufficiency (obviously gold standards for causation), the vast majority of experiments are done on cultured hippocampal slices (which likely affects various molecular mechanisms that may artefactually accentuate potential processes) and with highly non-physiological conditions (See specific comments for details). Crucially, a key missing element is demonstrating that an astrocytic-driven process detectably affects post synaptic neurons in non-cultured tissue when a physiologically realistic afferent stimulation protocol is used (See specific comments for further details). I emphasize, the approach used is elegant, but it is not without problems, and these need to be addressed.

More broadly, throughout, the study is underpowered with key experiments using only n of 3-6 (All of Fig. 1D, subsets of Fig. 1F, All of Fig. 1H, subsections of 2E, all of 2F, subsections of 3B, all of Fig. 4C, All of Fig. 4G, subsections of 5D, all of Suppl Fig 1, 2, 4, and 9), which while acceptable for some more established replications (e.g. AP5 blocks induction of LTD), is in my opinion below standard for the support required for such dogma changing conclusions. I emphasize, even such low n's might be acceptable if there was little overlap among experimental conditions, but if 3 of 5 observations completely overlap with 3 of 5 in another condition, the fact that n of 2 of 5 manages statistical significance, is nonetheless unlikely representative of the actual population, which is critical for such a study. A power analysis/analysis of normality/risk of false positives/etc. is warranted, and a reflective increase in n's throughout is important.

Specific Comments:

1) Overall, conceptually it is hard to envision how under realistic physiological conditions glutamate release from axons is not sufficient to elicit LTD directly, but glutamate mediated, neuron-triggered

glutamate release from astrocytes is sufficient? Of course it is possible, but given the higher density of glutamate vesicles in axon terminals, the higher concentration of glutamate in neuronal vesicles (due to higher concentrations in the cytoplasm, see detailed studies by Storm-Mathisen and colleagues), the closer proximity to post-synaptic receptors, and the smaller space of the synapse, its hard to envision the process whereby glutamate release from axons is able to stimulate astrocytes enough (but apparently not post synaptic neurons), that they release glutamate that is sufficient to trigger LTD. More specifically, it is hard to envision that stimulating a large portion of the entire afferent bundle (~70% based on initial stimulus intensity response curves, Page 35 methods) at 1Hz (1/SECOND) is not able to elicit plasticity directly, but the apparent resultant increase of 2 astrocytic Ca²⁺ spikes per MINUTE (Fig. 1C, right panel), and a consequent increase in astrocytic glutamate release that is only able to elicit in the post synaptic neuron, a single additional SIC per MINUTE (Fig. 1H) under highly non-physiological conditions (Zero Mg²⁺ plus added saturating glycine concentrations at the NMDAR), is enough to elicit LTD under physiologically realistic conditions?

Thus, it is important to consider what about the current study design might enable such outcomes artefactually? First, while the facility of cultured slices to implement the various molecular manipulations is clear, it remains possible that culture conditions alter molecular mechanisms such that astrocytic vesicular glutamate release becomes artefactually dominant (Fiacco et al. 2009 Ann. Rev. Pharmacol. Toxicol). Moreover, the authors are stimulating a large portion of the entire afferent bundle, which does not occur in vivo. Additionally, although it is not clear from the description in the methods, from the various figure diagrams, it looks as if the stimulating electrode or the location of light activation of Channelrhodopsin expressing astrocytes are quite close to the sites of recording (fluorescence or electrophysiology). Collectively, it seems likely that excessive/non-physiological stimulation protocol combined with potential culture condition amplification of glial signaling might explain the authors results, which although intriguing do not align very well with other well established factors (see above). Thus, key experiments need to be replicated in non-cultured slices and with more physiologically realistic parameters.

There are probably many ways to explore this crucial issue, and the more methods tested the better, but at the simplest level, it is crucial that the authors can observe some form of postsynaptic neural response (electrical or Ca²⁺) to the physiological LTD stimulation protocol (i.e. normal Mg²⁺, no added glycine, minimal stimulation protocol, or at least one that reflects in vivo activity) that is blocked by selective inhibition of the proposed key astrocytic mechanisms, P38a. This could be done using the authors in vivo transfection approach followed by preparation of acute slices, but a between slice study will likely make it hard to identify relevant responses, which are likely minimal. Thus, an easier approach would be to use acutely prepared slices, and use a broad spectrum p38 antagonist, bath applied. If the authors' hypothesized cascade is correct, there should be some observable acute post synaptic neuronal response to 1 Hz minimal afferent stimulation (with normal Mg²⁺ and no glycine) that is eliminated by blocking p38. This is fundamental. From the data presented it is fully possible that in the in vivo studies, astrocytic signaling, including p38, is permissive, but not causative, and that is a fundamentally different conclusion than the authors have made.

Beyond that specific critical experiment, generally, the astrocyte Ca²⁺ imaging experiments should be replicated in acute slices with more realistic stimulation protocols (lower intensity, and far enough away to preclude direct stimulation of astrocytes in the vicinity). Even if astrocyte Ca²⁺ does increase when the whole afferent bundle is stimulated, that does not mean it does in vivo or in response to more realistic afferent activity, and this is likely a key cause of contention in this field (Fiacco et al. IBID; Bazargani and Atwell 2016 Nat. Neurosci.). Moreover, given the gap junctional connections among astrocytes, even stimulation at greater distances might elicit waves of astrocytic Ca²⁺ that are not physiological, i.e. triggered by afferent activity as opposed to direct electrical excitation. Thus, distant stimulation-induced Ca²⁺ spiking should be demonstrated pharmacologically to be dependent

on glutamate release from axonal afferents.

2) Almost certainly related to the above, the broad conclusion raises questions about synaptic specificity, which the authors at least address to a degree in the discussion. But beyond this conceptual concern, in the control experiments displayed in Suppl Fig. 2B, given that the experimental stimulated pathway and the control, non-stimulated pathway electrodes appear to be at the same height within the hippocampal lamina, with one being upstream and one being downstream of the recorded cell, it is somewhat surprising that none of the manipulations used, induced plasticity in the “control, non-stimulated pathway”, given that Schaeffer collaterals typically traverse across the lamina and thus should share synapses among up- and down-stream stimulated, presumed common axons? Moreover, if the necessary and sufficient trigger is actually astrocytic Ca²⁺ spiking, which is widespread, why don't all synapses exhibit LTD?

3) The authors assume that their BAPTA-loading approach effectively buffers Ca²⁺ across all astrocytes loaded with co-injected dye. However, the authors did not quantify efficacy of BAPTA loading, which may not be the same as dye-loading, and like the image of dye loading may be graded across distant astrocytes. Regardless, the authors should directly test whether the BAPTA loading technique actually buffers stimulation induced astrocytic Ca²⁺ spiking, and if so in what percent of astrocytes at various locations. There is current controversy over local domains of Ca²⁺ signaling within astrocytes in the context of synaptic modulation, and determining if Ca²⁺ buffering in the soma and processes is effective is an important detail for the field.

4) Related to several points above, the authors state that LTD induction was diminished in IP3 receptor knockout mice, based on a p-value of 0.06 and N=13, but apparently without conducting an ANOVA across groups. Moreover, the authors do not test whether stimulus-induced Ca²⁺ spiking is abolished in the IP3 receptor knockout mice, which is the posited underlying mechanism. This is an important control generally, but especially in the context of concerns about whether the proximity of the stimulating electrode to the recording site might directly excite astrocytes (point 1, above).

Minor comments:

1) The authors indicate in the methods section that acute and cultured slices are used, and in some cases it is obvious, but generally, it is not always clear which type of slices are being used in the various sections and experiments. The authors should make it clear in the results section which type of slices are being used, acute or cultured, for each experiment.

2) In some sections of the results, the authors refer to their use of low frequency stimulation as LTD, even when electrophysiology is not conducted. If LTD has not been exhibited, it is inappropriate to refer to a stimulation protocol as LTD. This is true generally, but especially given that postsynaptic depolarization is necessary for LTD induction, which means that in fact it is unlikely that low frequency stimulation alone does induce LTD. Even if it can, if there is no quantification of LTD, there should be no suggestion of LTD in the protocol. All that is known is that low frequency stimulation occurred.

3) Typo in methods section: the last sentence of what sounds like the slice preparation description (“Slices were then transferred to...”) is appended at the end of the electrophysiology description, which seems out of place.

4) No P-values expressed in the text of the results for LTD with Bapta-loaded astros, P.5 4th to last line, and some other places, please double check throughout.

5) Some of the images that are designed to display specificity of loading/labeling/etc. are too non-

representative, too low a resolution, and completely lacking quantification to be of much value. For example, Fig. 1b, shows perhaps a single GCaMP6F loaded cell (not representative), the image is so small its difficult if not impossible to determine if it overlaps with blue and/or white (too small, poor resolution, poor choice of colors to detect overlap), and there is no quantification of the ratio of GFAP versus Neun cells that co-localize with GCaMP, or any description of how co-localization was thresholded. I emphasize that I do not doubt that there is some GCaMP in some astrocytes, but a more rigorous characterization is required if we are to be able to accurately interpret the resultant stimulus-induced fluorescent signals.

6) The authors should state more clearly how values for LTD expressed in the text relate to the time plots shown in the figures, is it the last point, the peak, the area under the curve? How was this analyzed statistically? In the stats chart it doesn't specify if they are repeated measure ANOVAs, which they should be, and there is no mention of interactions between factors, or consequent interpretation.

7) Do the authors have an explanation for why afferent stimulation induced LTD manifests quickly and then fades to steady state (but is not sufficient without astrocyte Ca²⁺ spiking), whereas astrocyte Ca²⁺ spiking alone is sufficient but slowly develops? This should be discussed.

8) So called surface staining and total fluorescence of EGFP tagged GluA2 subunits looks remarkably similar, and the surface staining does not look membrane delimited (i.e. there is no signal lacking core to spines or dendrites). Have the authors controlled for channel bleed through from EGFP, or other explanations for this apparent issue?

Reviewer #2 (Remarks to the Author):

This manuscript details a study determine the role of astrocytes in the induction of CA3 to CA1 hippocampal LTD. The study is well conducted and thoroughly examines each step of LTD induction including astrocyte calcium, vesicular release and postsynaptic glutamate action, revealing an unexpected role for astrocyte p38a MAPK. The study utilises a range of appropriate and innovative cutting-edge techniques and approaches from transgenic mouse models to optogenetics and targeted AAV transfection. Experiments are well conducted with overall appropriate controls. This study significantly advances our knowledge of the mechanisms of hippocampal LTD by elucidating the astrocyte mechanisms that are critical to its induction.

Comments:

1. Fig.3C. EGFP_GluA2 expression. There does not seem to be specific details in the methods section for these experiments. 3d appears to show ratio of spine fluorescence/dendrite fluorescence. How was this done? a specific area or cross-sectional line scan of spine and adjacent dendritic portion? If the graphs depict a ratio, what is the difference between "surface" and "total". Observation values are quoted as 116 spines n=3. Is this 3 dendrites, cells or slices? Control experiments with MK801 as in A,B would have strengthened link.

2. Fig5. Astro P38a MAPK is necessary for astrocyte–neuron communication. Data is shown from astrocytes expressing GluSNFR glutamate sensor, florescence increases are seen which further increase following LFS. Fluorescence increases are blocked in astrocyte p38a knock down slices. Recording SICs indicates that in p38a expressing slices there is no significant increase in SICs following LFS.

However, in the control experiments for Fig5d significance is reached with n=14 recordings while only 7 repeats are conducted in the astrocyte p38a knockdown slices. The data spread is large in both

conditions. It therefore raises the question whether an equal number of repeats would also show significance. Is this the case?

-It may also be that frequency is not the correct metric of SIC inhibition.

3. The statement that “p38a MAPK is required for glutamate release from astrocytes” is overenthusiastic and not completely supported by the data. Neither GluSNFR fluorescence events nor SICs are completely blocked by p38a knock-down (cf TeTX). It would be more correct to state that p38a is required for LFS induced astrocyte glutamate release (but see point addressing significance above).

4. Fig.6. Specific p38a deletion in astrocytes enhances long-term memory in vivo.

To determine a role for astrocyte p38a mediated LTD in memory the study shows data from experiments on fear conditioning. The aim seems to be to test the hypothesis that blocking LTD will inhibit the loss of a memory (ie reduce forgetting).

A study is cited (ref 69, Miguez et al) as precedence for this hypothesis which supports a role for LTD “forgetting” in an object location task, where animals “forget” in 10 days. Here LTD block by AMPAR manipulation resulted in a lack of “forgetting”(Miguez etal).

However, the contextual fear model used in this present study does not exhibit a similar “forgetting” profile and raises doubts about whether this is a suitable model paradigm for testing this LTD related hypothesis. The lack of freezing decay also makes interpretation difficult.

In this study mice (p38a lox/lox) are delivered foot shocks in a box and then 2 days later fear conditioning is quantified as % freezing following re-placing the mouse in the box. This happens to all mice then some are transfected with AAV5-Cre targeted at neurons or astrocytes and there is a vehicle group. At day 30, mice are again placed in the original box and % freezing measured. The data shows that all groups start at day 2 with ~ 40% freezing, after the 30 days the freezing in vehicle mice is not different. Therefore after 30 days the control vehicle mice have not “forgotten” the shock. In the p38a astrocyte knock-down animals the mice actually freeze more than at day 2. This therefore seems very different to not “forgetting”. The data for p38 knockdown animals in Fig. 6d do not therefore support a reduction in memory removal linked to LTD. The possibility that an increase in behavioral “freezing” is by another mechanism cannot be excluded. The direct correlation of freezing % to % transfected astrocytes shown in Fig. 6e that is used to support the ltd-memory hypothesis in fact might strongly support an alternative explanation, especially since (as shown in Fig.6e) that highest astrocyte transfection results in 80% freezing which is much higher than freezing seen at the beginning of the experiment at day 2. In addition to LTD block it is possible that p38a knockdown has other effects eg is it known what would be the effect of a transfected animal which is not exposed to shock then infected with an astrocyte p38a knockdown and freezing measured?

Other.

Wording in the manuscript should indicate that this is a study on astrocyte roles in hippocampal LTD. The study of Min and Nevian cited in the manuscript should also be more fully acknowledged as a study of spike timing dependent LTD in the cortex. This does not diminish the impact of the present study as it significantly extends knowledge, but brain region identification is important.

P.16.(and elsewhere) “astrocytes are obligatory intermediates for LTD expression add “in hippocampal CA1”.

Page 4 line 3. “contributes to retention of long term memory”. See point above

Page 5. 7 lines from bottom 50mM Ap5.

P6. para 3 l2. Exocytic or exocytotic?

P11. L7 from bottom. Place or site of action?

Check text throughout for p38a instead of alpha.

When say "infected/uninfected", would "transfected/untransfected" be more appropriate?

Include details on how the "kymograph" in 5d is generated. If from individual cells from slice on left, indicate which cells.

It is shown that astrocyte BAPTA infusion does not block LTP induction, how do the authors explain this observation in the context of Henneberger et al (Nature 2010) findings at the same synapse.

Reviewer #1

This manuscript presents experiments designed to test the role of astrocytic vesicular glutamate release in low frequency stimulation induced hippocampal LTD, and associated memory processing. The authors use an impressive array of electrophysiological, multiphoton fluorescence imaging, molecular manipulations and behavioral studies to test the necessity and sufficiency for hippocampal LTD of stimulation-induced astrocytic Ca²⁺ spiking and resultant vesicular glutamate release. The authors report the following:

- 1) In cultured hippocampal slices transfected with GCaMP6f under the GFP promotor, local low frequency (1Hz) electrical stimulation in the region of the Schaeffer collaterals increases the frequency of detectable Ca²⁺ spikes.*
- 2) Similar stimulation also elicited NMDA receptor-dependent slow inward currents (SICs) in patch-clamped CA1 pyramidal cells (all done in zero Mg²⁺ and added glycine).*
- 3) Local injection of BAPTA (20mM) into a single astrocyte, which resulted in presumed GAP-junction mediated spread to other astrocytes (evidenced by spread of fluorescent dye also included in the patch injector) reduced/prevented subsequent induction of LTD in a patch-clamped (-40mV) CA1 pyramid, as did bath application of the NMDA receptor antagonist, AP5, but not the mGluR antagonist MCPG.*
- 4) Such astrocyte BAPTA loading did not prevent induction of LTP, induced by 3Hz stimulation and stronger depolarization of the CA1 pyramid (0mV).*
- 5) None of the above manipulations affected transmission amplitude in an unpaired stimulated pathway.*
- 6) Induction of LTD was “diminished” in IP3-receptor knockout mice.*
- 7) Transfection of cultured hippocampal slices with GFAP driven tetanus toxin to prevent vesicular release reduced the frequency of spontaneous and stimulus-evoked SICs, and reduced/alterd the time course of LTD recorded in CA1 pyramids.*
- 8) Selective light stimulation of astrocytes, via photo-activation of channelrhodopsin transfected under the GFAP promoter depolarized astrocytes, and when repeated at low frequency increased Ca²⁺ spiking in astrocytes, SICs in CA1 pyramidal cells, and induced LTD in CA1 pyramidal cells (when clamped at -40mV during photostimulation), with the latter two events being blocked by AP5. Such LTD was also reduced, but not blocked by the NR2B selective antagonist Ifenprodil.*
- 9) Astrocyte photostimulation-induced LTD was not prevented by a cocktail of antagonists designed to block GABAergic, cannabinoid, cholinergic, purinergic and mGlu receptors, or by the addition of D-serine at concentrations expected to fully activate the NMDAR.*
- 10) Astrocyte photostimulation-induced LTD did not occur when the cell was not depolarized to -40mV, unless extracellular Mg²⁺ was removed, and it was blocked by postsynaptic MK-801 (applied in the patch pipette, which also blocked electrical stimulation induced LTD).*
- 11) Astrocyte photostimulation-induced a translocation of EGFP-tagged GluA2 subunits of AMPAR from spines to dendrites as expressed by the ratio of EGFP fluorescence in the two compartments, as well as by subsequent surface staining by antibodies under non-permeabilizing conditions.*
- 12) The authors manipulated cultured slices with a variety of molecular tools to delete P38 (a or b), known to be crucial for LTD induction, either globally or selectively in neurons or astrocytes, and only astrocyte selective P38a or broad spectrum antagonism significantly attenuated LTD.*
- 13) In vivo injections of viruses into various cre-lines that selectively diminished astrocytic p38a, but not selective ablation of neuronal p38a prevented subsequent induction of LTD in*

acutely prepared hippocampal slices, and enhanced the magnitude of freezing responses after 30 days after shock-induced conditioned fear, an effect that correlated with efficacy of astrocytic p38a knockdown.

Based on these observations, the authors conclude that “the action of astrocytic glutamate on postsynaptic NMDARs is not only a superimposed effect of astrocytes on excitatory synapses, with important physiological consequences of its own, but that this is the intrinsic mechanisms by which classical (low frequency-induced) NMDAR-dependent LTD operates.” Such a conclusion would obviously be revolutionary to our thinking about memory storage and the broad field of neuron-astrocyte signaling. Accordingly, the topic and results of this manuscript will be of widespread interest to the neuroscience community and readership of Nature Communications. Moreover, as touched on above, the authors have used an impressive array of tools to systematically test most of the proposed sequential mechanisms, all of which generally support their provocative conclusion.

However, as the authors acknowledge, this is currently a highly contentious issue generally, and to draw such radical conclusions requires exemplary experimental support. In this context, there are several substantive issues that lessen my enthusiasm for this manuscript, some of which can be directly experimentally tested, and thus should be.

General comments

1. In particular, while everything the authors tested shows correlation, necessity and sufficiency (obviously gold standards for causation), the vast majority of experiments are done on cultured hippocampal slices (which likely affects various molecular mechanisms that may artefactually accentuate potential processes) and with highly non-physiological conditions (see specific comments for details). Crucially, a key missing element is demonstrating that an astrocytic-driven process detectably affects post synaptic neurons in non-cultured tissue when a physiologically realistic afferent stimulation protocol is used (see specific comments for further details). I emphasize, the approach used is elegant, but it is not without problems, and these need to be addressed.

As described with more detail below, for the specific comments, we have now carried out new experiments or increased the ‘n’ of previous experiments to reproduce the most critical findings of this study using acute slices. These include:

- activation of astrocytes (Ca²⁺ spikes) and astrocyte-neuron communication (SICs) during low-frequency stimulation (LFS): Fig. 1c, d; Suppl. Fig. 1a, b.
- requirement of astrocytic Ca²⁺ signaling for LTD: Fig. 1f.
- sufficiency of photostimulation of astrocytes to induce LTD: Suppl. Fig. 7a, b.
- requirement of postsynaptic NMDA receptors for LTD induced by astrocytic photostimulation: Fig. 3a, b.
- requirement of astrocytic p38alpha for astrocyte-neuron communication (SICs) and for LTD: Fig. 5d-e; Suppl. Fig. 13.

In addition, we now have the key missing element pointed out by the reviewer: detection of a p38α-dependent postsynaptic response (depolarization) driven by astrocytes during LTD induction (Suppl. Fig. 11; see also comment 1, third part). This was also done with acute slices and mild electrical stimulation.

We appreciate the reviewer’s emphasis on this point. As a consequence, we can now safely say that most of the experimental data in this study were obtained with acute slices or in vivo.

2. *More broadly, throughout, the study is underpowered with key experiments using only n of 3-6 (all of Fig. 1D, subsets of Fig. 1F, all of Fig. 1H, subsections of 2E, all of 2F, subsections of 3B, all of Fig. 4C, all of Fig. 4G, subsections of 5D, all of Suppl Fig 1, 2, 4, and 9), which while acceptable for some more established replications (e.g. AP5 blocks induction of LTD), is in my opinion below standard for the support required for such dogma changing conclusions. I emphasize, even such low n's might be acceptable if there was little overlap among experimental conditions, but if 3 of 5 observations completely overlap with 3 of 5 in another condition, the fact that n of 2 of 5 manages statistical significance, is nonetheless unlikely representative of the actual population, which is critical for such a study. A power analysis/analysis of normality/risk of false positives/etc. is warranted, and a reflective increase in n's throughout is important.*

We have now increased the 'n' for most experiments (Figs. 1c, d, f, Fig. 3a, b, Fig. 5e, g and Suppl. Figs. 3b –previously 2– and 13), or in some cases, added data with acute slices rather than organotypic cultures (Suppl. Fig. 7a, b –related to Fig. 2e). All the corresponding statistics have been updated and are explicitly listed in the Supplementary Table. To note, all these new data correspond to experiments carried out with acute slices.

Specific comments

1. *Overall, conceptually it is hard to envision how under realistic physiological conditions glutamate release from axons is not sufficient to elicit LTD directly, but glutamate mediated, neuron-triggered glutamate release from astrocytes is sufficient? Of course it is possible, but given the higher density of glutamate vesicles in axon terminals, the higher concentration of glutamate in neuronal vesicles (due to higher concentrations in the cytoplasm, see detailed studies by Storm-Mathisen and colleagues), the closer proximity to post-synaptic receptors, and the smaller space of the synapse, its hard to envision the process whereby glutamate release from axons is able to stimulate astrocytes enough (but apparently not post synaptic neurons), that they release glutamate that is sufficient to trigger LTD. More specifically, it is hard to envision that stimulating a large portion of the entire afferent bundle (~70% based on initial stimulus intensity response curves, Page 35 methods) at 1Hz (1/SECOND) is not able to elicit plasticity directly, but the apparent resultant increase of 2 astrocytic Ca²⁺ spikes per MINUTE (Fig. 1C, right panel), and a consequent increase in astrocytic glutamate release that is only able to elicit in the post synaptic neuron, a single additional SIC per MINUTE (Fig. 1H) under highly non-physiological conditions (Zero Mg²⁺ plus added saturating glycine concentrations at the NMDAR), is enough to elicit LTD under physiologically realistic conditions?*

We understand the skepticism of the reviewer with respect to the ability to induce LTD by the sparse activation of NMDA receptors elicited by astrocytic glutamate release, as compared to the regular repetitive release provided by the presynaptic neuron. To directly address the biophysical plausibility of our model, we have calculated the charge transfer associated to SICs versus EPSCs during a typical LTD induction protocol. As the reviewer notes, the frequency of these events is very different (1-2 events per minute for SICs, versus 1 event per second -1 Hz- for EPSCs). However, the other relevant parameter is the duration of each event. As shown in Suppl. Fig. 2a, SICs have much longer durations (longer onset and offset times) than EPSCs. This will imply that, even if the amplitudes are not very different, the amount of charge transferred in a SIC event will be much larger than that of an EPSC. We have explicitly calculated this parameter from the areas of SIC and EPSC events (red and blue shaded areas, in Suppl. Fig. 2b). As plotted in Suppl. Fig. 2c, the average charge transfer for each EPSC during

LTD induction is 5.0 ± 2.3 pA*s, versus 376 ± 154 pA*s for each SIC. If we take into account that there will be 300 EPSCs during the 5 min of LTD induction (1 Hz), and about 5 SICs (one per minute), this will give a total charge transfer of about 1500 pA*s for EPSCs and about 1880 pA*s for SICs. Therefore, even within the variability of these numbers, we can conservatively conclude that NMDA receptors from the postsynaptic neuron will be exposed to similar amounts of glutamate from the presynaptic terminal as from neighboring astrocytes during LTD induction. Admittedly, most glutamate released from the presynaptic neuron will be concentrated on postsynaptic receptors, whereas astrocytic glutamate may spread over a larger area. However, the relative relevance of synaptic versus extrasynaptic NMDA receptors for eliciting LTD is still an unresolved issue, as well as the locality of astrocytic glutamate release. These considerations are now presented in the Results (beginning of page 5) and Discussion (page 19 and beginning of page 20) of the revised manuscript.

On the other hand, we do not believe that presynaptic glutamate is completely ineffective to induce synaptic depression. We would rather propose that the astrocyte and the presynaptic neuron contribute to different phases of LTD. This interpretation is further elaborated below (minor comment 7) and is now explicitly considered in the Discussion of the revised manuscript (page 19 and beginning of page 20).

- *Thus, it is important to consider what about the current study design might enable such outcomes artifactually? First, while the facility of cultured slices to implement the various molecular manipulations is clear, it remains possible that culture conditions alter molecular mechanisms such that astrocytic vesicular glutamate release becomes artifactually dominant (Fiacco et al. 2009 Ann. Rev. Pharmacol. Toxicol). Moreover, the authors are stimulating a large portion of the entire afferent bundle, which does not occur in vivo. Additionally, although it is not clear from the description in the methods, from the various figure diagrams, it looks as if the stimulating electrode or the location of light activation of Channelrhodopsin expressing astrocytes are quite close to the sites of recording (fluorescence or electrophysiology). Collectively, it seems likely that excessive/non-physiological stimulation protocol combined with potential culture condition amplification of glial signaling might explain the authors results, which although intriguing do not align very well with other well established factors (see above). Thus, key experiments need to be replicated in non-cultured slices and with more physiologically realistic parameters.*

As mentioned above, we have now repeated most critical experiments using acute slices, to avoid potential artifacts derived from the use of cultured slices. In addition, we have now evaluated how the distance between the stimulation electrode and the recorded cell affects Ca^{2+} spikes in the astrocyte, using acute slices and mild electrical stimulation (theta capillaries, 4-9 μm tip). As shown in Suppl. Fig. 1, a, b (page 4, second paragraph), stimulation-induced Ca^{2+} spikes are detectable over 150 μm away from the stimulation electrode.

On the other hand, we also want to clarify that for most of our experiments, we are far from stimulating the majority of the afferents. Unfortunately, this interpretation comes from a misstatement in the Methods section that was meant to refer to field recordings and was also inadvertently applied to whole-cell recordings. This is now corrected in the revised manuscript (page 44, first and second paragraphs). Indeed, for our field recordings (used exclusively for Fig. 4d, e), we selected a stimulation strength yielding about 70% of the maximal response. However, the overwhelming majority of our electrophysiological recordings were carried out under whole-cell configuration. For these experiments, we aimed at obtaining a response of 50-100 pA in

amplitude. The size of this EPSC would roughly correspond to the activation of 4-to-10 synapses (considering an average miniature EPSC amplitude of 10-15 pA).

- *There are probably many ways to explore this crucial issue, and the more methods tested the better, but at the simplest level, it is crucial that the authors can observe some form of postsynaptic neural response (electrical or Ca²⁺) to the physiological LTD stimulation protocol (i.e. normal Mg²⁺, no added glycine, minimal stimulation protocol, or at least one that reflects in vivo activity) that is blocked by selective inhibition of the proposed key astrocytic mechanisms, P38a. This could be done using the authors in vivo transfection approach followed by preparation of acute slices, but a between slice study will likely make it hard to identify relevant responses, which are likely minimal. Thus, an easier approach would be to use acutely prepared slices, and use a broad spectrum p38 antagonist, bath applied. If the authors' hypothesized cascade is correct, there should be some observable acute post synaptic neuronal response to 1 Hz minimal afferent stimulation (with normal Mg²⁺ and no glycine) that is eliminated by blocking p38. This is fundamental. From the data presented it is fully possible that in the in vivo studies, astrocytic signaling, including p38, is permissive, but not causative, and that is a fundamentally different conclusion than the authors have made.*

As requested by the reviewer, we now present an acute postsynaptic response induced by astrocytic activation during LTD induction, under physiological conditions (acute slices, normal Mg²⁺, no glycine, mild stimulation). Specifically, we have observed that, under current clamp configuration, the postsynaptic neuron undergoes large subthreshold depolarizations during LTD induction (Suppl. Fig. 11; page 14, second paragraph). These responses are very slow (similar to SICs) and sporadic (mean frequency of 1.5 min⁻¹), in clear contrast with the regular and fast synaptic responses timed with the 1 Hz presynaptic stimulation (see inset in Suppl. Fig. 11a). These properties are consistent with being driven by astrocytic activity. Importantly, these slow depolarizations are strongly attenuated when blocking p38 or NMDA receptor activity (SB203580 and AP5, respectively).

- *Beyond that specific critical experiment, generally, the astrocyte Ca²⁺ imaging experiments should be replicated in acute slices with more realistic stimulation protocols (lower intensity, and far enough away to preclude direct stimulation of astrocytes in the vicinity). Even if astrocyte Ca²⁺ does increase when the whole afferent bundle is stimulated, that does not mean it does in vivo or in response to more realistic afferent activity, and this is likely a key cause of contention in this field (Fiacco et al. IBID; Bazargani and Atwell 2016 Nat. Neurosci.). Moreover, given the gap junctional connections among astrocytes, even stimulation at greater distances might elicit waves of astrocytic Ca²⁺ that are not physiological, i.e. triggered by afferent activity as opposed to direct electrical excitation. Thus, distant stimulation-induced Ca²⁺ spiking should be demonstrated pharmacologically to be dependent on glutamate release from axonal afferents.*

We have now evaluated the frequency of Ca²⁺ spiking as a function of the distance to the stimulation electrode, using acute slices and mild stimulation (glass electrodes). As mentioned above (new Suppl. Fig. 1a, b), we detect an increase in Ca²⁺ spiking during LFS with the stimulation electrode up to 150-200 μm away from the imaged astrocytes (page 4, second paragraph). Therefore, even though no *ex-vivo* experiment is truly physiological, we believe we are mimicking the standard conditions used by most laboratories studying LFS-induced LTD.

- 2. Almost certainly related to the above, the broad conclusion raises questions about synaptic specificity, which the authors at least address to a degree in the discussion. But beyond this conceptual concern, in the control experiments displayed in Suppl Fig. 2B, given that the experimental stimulated pathway and the control, non-stimulated pathway electrodes appear to be at the same height within the hippocampal lamina, with one being upstream and one being downstream of the recorded cell, it is somewhat surprising that none of the manipulations used, induced plasticity in the “control, non-stimulated pathway”, given that Schaeffer collaterals typically traverse across the lamina and thus should share synapses among up- and down-stream stimulated, presumed common axons? Moreover, if the necessary and sufficient trigger is actually astrocytic Ca²⁺ spiking, which is widespread, why don't all synapses exhibit LTD?*

In order to provide a more realistic image of the location of the stimulating electrodes in our experiments, we now include a representative picture of the actual configuration for electrophysiological recordings, with the stimulating and recording electrodes on an acute slice (Suppl. Fig. 3a, in the revised version). As discussed above (comment 1, second part), our stimulation protocols are probably activating just a few synapses, and therefore, it is not unlikely that their corresponding afferents do not overlap. Nevertheless, for a fraction of the experiments (about 15 %) we did observe synaptic depression of both stimulated and control (non-stimulated) pathway. These experiments were excluded from the analysis, because we could not be certain about the independence of the stimulating pathways.

- 3. The authors assume that their BAPTA-loading approach effectively buffers Ca²⁺ across all astrocytes loaded with co-injected dye. However, the authors did not quantify efficacy of BAPTA loading, which may not be the same as dye-loading, and like the image of dye loading may be graded across distant astrocytes. Regardless, the authors should directly test whether the BAPTA loading technique actually buffers stimulation induced astrocytic Ca²⁺ spiking, and if so in what percent of astrocytes at various locations. There is current controversy over local domains of Ca²⁺ signaling within astrocytes in the context of synaptic modulation, and determining if Ca²⁺ buffering in the soma and processes is effective is an important detail for the field.*

As requested by the reviewer, we have now directly evaluated the effectiveness of our BAPTA-loading method to quench Ca²⁺ signals in astrocytes. As shown in new Suppl. Fig. 1b, Ca²⁺ spiking was virtually abolished when astrocytes were loaded with BAPTA. This occurred at all distances from the stimulating electrode. As pointed out by the reviewer, there is a current controversy on the specific roles of local domains of Ca²⁺ within the astrocyte, separate from somatic Ca²⁺. In this study we are only monitoring somatic Ca²⁺, but we believe it is not necessary to study local Ca²⁺ domains in this case, since our BAPTA-loading method is already blocking LTD expression.

- 4. Related to several points above, the authors state that LTD induction was diminished in IP3 receptor knockout mice, based on a p-value of 0.06 and N=13, but apparently without conducting an ANOVA across groups. Moreover, the authors do not test whether stimulus-induced Ca²⁺ spiking is abolished in the IP3 receptor knockout mice, which is the posited underlying mechanism. This is an important control generally, but especially in the context of concerns about whether the proximity of the stimulating electrode to the recording site might directly excite astrocytes (point 1, above).*

We apologize for the confusion about the statistical tests we have used. Indeed, we conducted ANOVA across groups for all experiments when comparing more than two conditions. Only when statistical significance was found across the group, a post-hoc test was used for pairwise comparisons. This is now explicitly stated in the Methods section (“Statistical analysis”). Also, all statistical analysis is described in each figure legend, and listed in the Suppl. Table. With respect to the IP3R2 knockout and the blockade of Ca²⁺ signals in these animals, it is also a controversial issue. It has been published that astrocytic Ca²⁺ signals are absent (or strongly reduced) in the soma (Petrovicz et al, J Neurosci 2008), but they persist in local microdomains (Srinivasan et al, Nat Neurosci 2015). The issue of the functionality of these local Ca²⁺ microdomains is indeed very interesting, but we believe it is outside the scope of our current study, because we find that quenching somatic Ca²⁺ (with BAPTA) blocks LTD.

Minor comments

1. *The authors indicate in the methods section that acute and cultured slices are used, and in some cases it is obvious, but generally, it is not always clear which type of slices are being used in the various sections and experiments. The authors should make it clear in the results section which type of slices are being used, acute or cultured, for each experiment.*

We apologize for this ambiguity. The type of slices is now explicitly declared in the Results section and in each figure legend of the revised manuscript. As a quick summary, acute slices were used for experiments in Figs. 1a-f, 3a-b, 4a-e, 5f-g, and Supplementary Figs. 1, 2, 3, 4, 6, 7, 11 and 13. As stated above, this now constitutes the majority of the slice work.

2. *In some sections of the results, the authors refer to their use of low frequency stimulation as LTD, even when electrophysiology is not conducted. If LTD has not been exhibited, it is inappropriate to refer to a stimulation protocol as LTD. This is true generally, but especially given that postsynaptic depolarization is necessary for LTD induction, which means that in fact it is unlikely that low frequency stimulation alone does induce LTD. Even if it can, if there is no quantification of LTD, there should be no suggestion of LTD in the protocol. All that is known is that low frequency stimulation occurred.*

As requested by the reviewer, we now refer to low-frequency stimulation (LFS) when we are describing the type of stimulation we are delivering, without the knowledge of the outcome for synaptic transmission.

3. *Typo in methods section: the last sentence of what sounds like the slice preparation description (“Slices were then transferred to...”) is appended at the end of the electrophysiology description, which seems out of place.*

Corrected (page 42).

4. *No P-values expressed in the text of the results for LTD with Bapta-loaded astros, P.5 4th to last line, and some other places, please double check throughout.*

‘P’ values are now explicitly declared for all results in the corresponding figure legends.

5. *Some of the images that are designed to display specificity of loading/labeling/etc. are too non-representative, too low a resolution, and completely lacking quantification to be of much value. For example, Fig. 1b, shows perhaps a single GCaMP6F loaded cell (not representative), the image is so small its difficult if not impossible to determine if it overlaps with blue and/or white (too small, poor resolution, poor choice of colors to detect overlap), and there is no quantification of the ratio of GFAP versus Neun cells that co-localize with*

GCaMP, or any description of how co-localization was thresholded. I emphasize that I do not doubt that there is some GCaMP in some astrocytes, but a more rigorous characterization is required if we are to be able to accurately interpret the resultant stimulus-induced fluorescent signals.

We apologize for the poor quality of the micrographs as result of data compression of the previous version. We have now changed the representative images for Figure 1b to better illustrate colocalization between GCaMP6f and GFAP staining. Furthermore, we have quantified this colocalization and found that close to 90% of the GCaMP6f-expressing cells are GFAP positive, ratifying the specificity of the astrocytic expression of GCaMP6f.

6. *The authors should state more clearly how values for LTD expressed in the text relate to the time plots shown in the figures, is it the last point, the peak, the area under the curve? How was this analyzed statistically? In the stats chart it doesn't specify if they are repeated measure ANOVAs, which they should be, and there is no mention of interactions between factors, or consequent interpretation.*

We apologize for the incomplete description of the statistical tests. This is now corrected, together with explicit statement of the time window used for comparing across different conditions in LTD experiments (in all cases, last 4.5 min of the time course). Synaptic plasticity changes were determined as the normalized change in average response size during the last 4.5 min of recording (41.5–45 min after induction protocol) compared to 6 min baseline. To illustrate the time course synaptic parameters were grouped in 1.5-min bins (fourth paragraph in page 44).

7. *Do the authors have an explanation for why afferent stimulation induced LTD manifests quickly and then fades to steady state (but is not sufficient without astrocyte Ca²⁺ spiking), whereas astrocyte Ca²⁺ spiking alone is sufficient but slowly develops? This should be discussed.*

This is indeed an interesting observation that had not escaped our attention. As described by the reviewer, it appears as if the two sources of glutamate (neuronal versus astrocytic) define two distinct phases of LTD expression. In this scenario, the fast-paced release of presynaptic glutamate would be responsible for the initial synaptic depression, which on its own would not be enough to maintain long-term depression. Complementarily, the slower release of astrocytic glutamate would fail to elicit fast depression, but would be needed to ensure the persistence of depression initiated by the presynaptic neuron. In this manner, both neuronal and astrocytic glutamate would contribute to the “canonical” time course of LTD expression. We had not

introduced these ideas originally in the Discussion section of the manuscript because of their speculative nature, but we are happy to do it now in the revised manuscript (end of page 19, beginning of page 20).

8. *So called surface staining and total fluorescence of EGFP tagged GluA2 subunits looks remarkably similar, and the surface staining does not look membrane delimited (i.e. there is no signal lacking core to spines or dendrites). Have the authors controlled for channel bleed through from EGFP, or other explanations for this apparent issue?*

The lack of membrane-delimited staining in our surface staining in Fig. 3c is due to the small size of the structures being imaged (in the order of 1-1.5 μm). This size is at the limit of the Z-resolution we used to image these rather dim signals. We are now providing low magnification images of the same samples, where the membrane-delimited staining can be easily observed in soma and thick dendrites (new Suppl. Fig. 8). In fact, the almost complementary fluorescence pattern of total and surface receptor that can be observed in these images is, in our opinion, a good testimony for the lack of significant signal bleed across channels.

Reviewer #2

This manuscript details a study determine the role of astrocytes in the induction of CA3 to CA1 hippocampal LTD. The study is well conducted and thoroughly examines each step of LTD induction including astrocyte calcium, vesicular release and postsynaptic glutamate action, revealing an unexpected role for astrocyte p38a MAPK. The study utilises a range of appropriate and innovative cutting-edge techniques and approaches from transgenic mouse models to optogenetics and targeted AAV transfection. Experiments are well conducted with overall appropriate controls. This study significantly advances our knowledge of the mechanisms of hippocampal LTD by elucidating the astrocyte mechanisms that are critical to its induction.

Specific comments

1. *Fig.3C. EGFP_GluA2 expression. There does not seem to be specific details in the methods section for these experiments. 3d appears to show ratio of spine fluorescence/dendrite fluorescence. How was this done? a specific area or cross-sectional line scan of spine and adjacent dendritic portion? If the graphs depict a ratio, what is the difference between “surface” and “total”. Observation values are quoted as 116 spines n=3. Is this 3 dendrites, cells or slices? Control experiments with MK801 as in A,B would have strengthened link.*

We apologize for the incomplete description of this experiment. The experimental details are now explicitly described in the Methods section of the revised manuscript (page 49, second paragraph). Specifically, regions of interest of the same size were defined around each spine and its adjacent dendritic shaft. It is this ratio (spine/dendrite) what is being plotted, for both the surface staining and the GFP channel (total receptor). As for the ‘n’, it corresponds to the number of slices, but in these experiments, only one neuron (and only one dendrite per neuron) was imaged in each experiment.

As for the effect of MK801, indeed we expect that it will block the effects of astrocytic activation on receptor redistribution. However, this experiment was meant to demonstrate that our protocol for astrocytic photoactivation produces changes in postsynaptic receptor trafficking, as it has been previously described for “canonical” LTD. We did not intend to further explore the problem of AMPA receptor trafficking in response to NMDA receptor activation, which has been extensively studied in the context of LTD.

2. *Fig5. Astro P38a MAPK is necessary for astrocyte–neuron communication. Data is shown from astrocytes expressing GluSNFR glutamate sensor, fluorescence increases are seen which further increase following LFS. Fluorescence increases are blocked in astrocyte p38a knock down slices. Recording SICs indicates that in p38a expressing slices there is no significant increase in SICs following LFS. However, in the control experiments for Fig5d significance is reached with n=14 recordings while only 7 repeats are conducted in the astrocyte p38a knockdown slices. The data spread is large in both conditions. It therefore raises the question whether an equal number of repeats would also show significance. Is this the case? It may also be that frequency is not the correct metric of SIC inhibition.*

As requested by the reviewer, we have increased the ‘n’ for both glutamate imaging (Fig. 5e) and SIC recordings (Fig. 5g). I believe we can now safely conclude that there is no significant increase in glutamate release or astrocyte-to-neuron communication when removing astrocytic p38alpha.

3. *The statement that “p38a MAPK is required for glutamate release from astrocytes” is overenthusiastic and not completely supported by the data. Neither GluSNFR fluorescence events nor SICs are completely blocked by p38a knock-down (cf TeTX). It would be more correct to state that p38a is required for LFS induced astrocyte glutamate release (but see point addressing significance above).*

We completely agree with the reviewer, and the corresponding statement has been modified accordingly (last two sentences in the Abstract).

4. *Fig.6. Specific p38a deletion in astrocytes enhances long-term memory in vivo. To determine a role for astrocyte p38a mediated LTD in memory the study shows data from experiments on fear conditioning. The aim seems to be to test the hypothesis that blocking LTD will inhibit the loss of a memory (ie reduce forgetting). A study is cited (ref 69, Miguez et al) as precedence for this hypothesis which supports a role for LTD “forgetting” in an object location task, where animals “forget” in 10 days. Here LTD block by AMPAR manipulation resulted in a lack of “forgetting”(Miguez et al). However, the contextual fear model used in this present study does not exhibit a similar “forgetting” profile and raises doubts about whether this is a suitable model paradigm for testing this LTD related hypothesis. The lack of freezing decay also makes interpretation difficult. In this study mice (p38a lox/lox) are delivered foot shocks in a box and then 2 days later fear conditioning is quantified as % freezing following re-placing the mouse in the box. This happens to all mice then some are transfected with AAV5-Cre targeted at neurons or astrocytes and there is a vehicle group. At day 30, mice are again placed in the original box and % freezing measured. The data shows that all groups start at day 2 with ~ 40% freezing, after the 30 days the freezing in vehicle mice is not different. Therefore after 30 days the control vehicle mice have not “forgotten” the shock. In the p38a astrocyte knock-down animals the mice actually freeze more than at day 2. This therefore seems very different to not “forgetting”. The data for p38 knockdown animals in Fig. 6d do not therefore support a reduction in memory removal linked to LTD. The possibility that an increase in behavioral “freezing” is by another mechanism cannot be excluded. The direct correlation of freezing % to % transfected astrocytes shown in Fig. 6e that is used to support the ltd-memory hypothesis in fact might strongly support an alternative explanation, especially since (as shown in Fig.6e) that highest astrocyte transfection results in 80% freezing which is much higher than freezing seen at the beginning of the experiment at day 2. In addition to LTD block it is possible that p38a knockdown has*

other effects eg is it known what would be the effect of a transfected animal which is not exposed to shock then infected with an astrocyte p38a knockdown and freezing measured?

As prompted by the reviewer, we have now tested other potential behavioural effects of removing p38alpha, which may be confounding the results for freezing behaviour after fear conditioning. For example, the increase in freezing observed in p38alpha knockdown animals could be due to changes in exploratory activity or anxiety behaviour. As shown in Suppl. Fig. 14c-f, animals with p38alpha removed from neurons or astrocytes were no different from controls (vehicle-injected) with respect to their global activity, grooming or spontaneous freezing behaviour, when they were not exposed to an electric shock. These animals were also tested in the open field. As shown in Suppl. Fig. 15, genetic deletion of p38alpha from astrocytes or neurons did not alter the distance travelled or average speed of the animals, nor their global activity levels. Importantly, there was no change either in their preference for periphery or the central area of the open field, suggesting that there were no changes in their anxiety levels. Although we cannot rule out more subtle effects, p38alpha removal does not appear to produce an enhancement of freezing behaviour in the absence of fear conditioning.

Nevertheless, we agree with the reviewer in that our previous interpretation of the effect of astrocytic deletion of p38alpha was overstated, since we could not detect memory loss in control animals over a period of 30 days. Therefore, all references to “reduced forgetting” or “enhanced memory retention/persistence” have now been removed throughout the text. These changes also affect the Abstract and Title of the manuscript. Nevertheless, taken at face value, our data do suggest an increase in long-term memory in animals without astrocytic p38alpha. We do not have a definitive explanation for this result (and this is now reflected in the Discussion of the revised manuscript), but we now hypothesize that it may be related to a potential role of LTD in memory generalization and flexibility. This is based on previous reports that memory generalization leads to an enhancement of fear responses over time (Wiltgen and Silva, *Learn Mem* 14, 313-317, 2007; Poulos et al, *Learn Mem* 23, 379-385, 2016; Pollack et al, *Learn Mem* 25, 298-308, 2018). This is considered a maladaptive process, which would be physiologically limited by new learning and memory flexibility (Richards and Frankland, *Neuron* 94, 1071-1084, 2017). Interestingly, LTD has been proposed to be required precisely for behavioural flexibility (Nicholls et al, *Neuron* 58, 104-117, 2008; Kim et al, *Nat Neurosci* 14, 1447-1454, 2011; Mills et al, *PNAS* 111, 8631-8636, 2014). Therefore, the enhancement of long-term fear memory in the astrocytic p38alpha-deleted animals may be a consequence of an impaired mechanism to prevent memory run-up due to overgeneralization. However, we should point out that our study does not intend to ascertain the cognitive function of LTD, and consequently, we have not directly tested behavioural flexibility or memory interference with our p38-deleted animals. Nevertheless, we believe the published literature justifies this speculative interpretation for our behavioural data in the revised manuscript (end of page 20 and beginning of page 21).

Other comments

- *Wording in the manuscript should indicate that this is a study on astrocyte roles in hippocampal LTD. The study of Min and Nevian cited in the manuscript should also be more fully acknowledged as a study of spike timing dependent LTD in the cortex. This does not diminish the impact of the present study as it significantly extends knowledge, but brain region identification is important.*

We now explicitly acknowledge that this study specifically refers to hippocampal LTD induced by low-frequency stimulation. The specific differences with the study by Min and

Nevian (2012) on spike timing-dependent LTD at cortical synapses is now more clearly stated (end of page 17).

- *P.16.(and elsewhere) “astrocytes are obligatory intermediates for LTD expression add “in hippocampal CA1”.*

Corrected (beginning of page 17, and other places throughout the text).

- Page 4 line 3. “contributes to retention of long term memory”. See point above.

As pointed out by the reviewer (comment 4), we cannot really conclude that deletion of astrocytic p38alpha prevents memory decay or enhances memory retention, because we do not detect spontaneous time-dependent memory loss in our assay. Nevertheless, our additional behavioral experiments (Suppl. Figs. 14 and 15) suggest that deletion of astrocytic p38alpha is truly acting on long-term memory of fear conditioning, rather than general (or non-specific) freezing behavior. Therefore, we have changed our previous statements of “memory retention” or “memory decay” to “modulation of long-term memory”. This term is more descriptive and less prone to over-interpretation. As mentioned above, these changes also apply to the Abstract and Title of the manuscript.

- *Page 5. 7 lines from bottom 50mM Ap5.*

Corrected.

- *P6.para 3 l2. Exocytic or exocytotic?*

We used “exocytic” as the adjective for “exocytosis”, as “endocytic” is used for “exocytosis”. But we will agree with the linguistic policy of the journal on this matter.

- *P11. L7 from bottom. Place or site of action?*

We used “place of action” because we referred to the cell-type where p38alpha is acting, rather than to its specific “site of action”. However, we recognize this is a rather ambiguous expression, and we will be happy to change it according to editorial policies.

- *Check text throughout for p38a instead of alpha.*

Corrected.

- *When say “infected/uninfected”, would “transfected/untransfected” be more appropriate?*

In general, we prefer to use “infection” when referring to a viral vector, and “transfection” when using other non-specific gene-delivery method. But again, we will be happy to comply with the editorial policies of the journal.

- *Include details on how the “kymograph” in 5d is generated. If from individual cells from slice on left, indicate which cells.*

Images were acquired at 1 Hz frame rate for 15 min. Minor drift in the XY plane of image stacks was post hoc corrected using TurboReg (ImageJ plugin). For kymograph analysis, ROIs were designed using the *Grid* plugin of ImageJ (44 x 44 px, equivalent to 14-25 x 14-25 μ m). The values for these ROIs are then analyzed using the *GECIquant* open source plugin for Fiji, to detect and analyze glutamate signals in astrocytes expressing genetically encoded iGluSnFr. ROIs pooled from two different experiments were used in Figure 5d. (Pag 46).

- *It is shown that astrocyte BAPTA infusion does not block LTP induction, how do the authors explain this observation in the context of Henneberger et al (Nature 2010) findings at the same synapse.*

We believe this may be due to the different strategies used for Ca²⁺ quenching and/or the differences in the LTP induction protocols. Thus, we used BAPTA loading, while Henneberger et al. used a Ca²⁺ clamping approach based on an EGTA/Ca²⁺ mix. These authors have shown that the Ca²⁺ clamping strategy is indeed more efficient in preventing Ca²⁺ rise induced by high-frequency stimulation. But this is also an important methodological difference: we induced LTP under voltage-clamp configuration, by pairing postsynaptic depolarization and only moderate (3 Hz) presynaptic stimulation. In contrast, Henneberger et al carried out field recordings, and employed high-frequency stimulation for LTP induction. This is a relevant point, since LTP expression has been shown to be less dependent on D-serine (or glycine) when facilitated by postsynaptic depolarization (Krasteniakov et al, Eur J Neurosci 21, 2782-2792, 2005). Therefore, the different requirement of astrocytic Ca²⁺ for LTP may just reflect the varying dependence on D-Serine for LTP under different induction conditions. These possibilities are now mentioned in the revised manuscript (first paragraph of page 18).

REVIEWERS' COMMENTS:

Reviewer #1 (Remarks to the Author):

The authors have very thoroughly addressed my concerns, making this manuscript clearer more sound. It is a very interesting and likely concept shifting piece of work, elegantly conducted using an amazing array of techniques, to really hone in on the concepts from multiple complementary angles.

As may have been evident, because of the complexity of this manuscript and ongoing controversy in the field generally, I spent a fair amount of time and effort on my review. Consequently, I wish to thank the authors for taking my comments seriously, and carefully addressing them. Although such thorough follow ups can be daunting, I do think it is important for the field to do it anyway.

Nice work all around.

Reviewer #2 (Remarks to the Author):

The points raised in the review have been addressed. A few minor issues remain:

Original Point 3: Although authors have changed some sentences in the manuscript to correct statements that suggest that p38a is necessary for astrocyte glutamate release, and not as shown in the study the *increase* in release following LFS, there are still some instances where this needs to be corrected eg

Page 13 – subsection title “p38a mapk is required for glutamate release from astro and astro neuron communication” .

Fig 5 title.” P38a mapk is necessary for astrocyte neuron communication”

minor

Abstract.line 1 “NMDA receptor (dependent?mediated?) long term...”

Supplementary figure legend 14. Line 8. “...were trained similarly than mice of figure 6”

Probably should be “were similarly trained to mice in figure 6”

Reviewer #2

The points raised in the review have been addressed. A few minor issues remain:

- *Original Point 3: Although authors have changed some sentences in the manuscript to correct statements that suggest that p38 α is necessary for astrocyte glutamate release, and not as shown in the study the increase in release following LFS, there are still some instances where this needs to be corrected eg*

Page 13 – subsection title “p38 α mapk is required for glutamate release from astro and astro neuron communication” .

Fig 5 title.” P38 α mapk is necessary for astrocyte neuron communication”

These statements have been reworded as “p38 α MAPK is required for the increase in astrocytic glutamate release after LFS” (subsection title, page 12) and “Astrocytic p38 α is necessary for LFS-induced astrocyte-to-neuron communication” (title for new Fig. 8).

- *Minor*

Abstract.line 1 “NMDA receptor (dependent?mediated?) long term...”

Corrected as “NMDA receptor-dependent”

*Supplementary figure legend 14. Line 8. “...were trained similarly than mice of figure 6”
Probably should be “were similarly trained to mice in figure 6”*

Corrected as indicated by the reviewer.